# Transitivity Meets Cyclicity: Explicit Preference Decomposition for Dynamic Large Language Model Alignment

**Yucong Huang** [1]   **Xiucheng Li** [1]   **Kaiqi Zhao** [1]   **Jing Li** [✉] [1]

## Abstract

Standard RLHF relies on transitive scalar rewards, failing to capture the cyclic nature of human preferences. While some approaches like the General Preference Model (GPM) address this, we identify a theoretical limitation: their implicit formulation entangles hierarchy with cyclicity, failing to guarantee dominant solutions. To address this, we propose the Hybrid Reward-Cyclic (HRC) model, which utilizes game-theoretic decomposition to explicitly disentangle preferences into orthogonal transitive (scalar) and cyclic (vector) components. Complementing this, we introduce Dynamic Self-Play Preference Optimization (DSPPO), which treats alignment as a time-varying game to progressively guide the policy toward the Nash equilibrium. Synthetic data experiments further validate HRC's structural superiority in mixed transitive–cyclic settings, where HRC converges faster and achieves higher accuracy than GPM. Experiments on RewardBench 2 demonstrate that HRC consistently improves over both BT and GPM baselines (e.g., +1.23% on Gemma-2B-it). In particular, its superior performance in the Ties domain empirically validates the model's robustness in handling complex, non-strict preferences. Extensive downstream evaluations on AlpacaEval 2.0, Arena-Hard-v0.1, and MT-Bench confirm the efficacy of our framework. Notably, when using Gemma-2B-it as the base preference model, HRC+DSPPO achieves a peak length-controlled win-rate of 44.75% on AlpacaEval 2.0 and 46.8% on Arena-Hard-v0.1, significantly outperforming SPPO baselines trained with BT or GPM. Our code is publicly available at https://github.com/lab-klc/Hybrid-Reward-Cyclic.

[1]Harbin Institute of Technology, Shenzhen, China. Correspondence to: Jing Li <jingli.phd@hotmail.com>.

*Proceedings of the 43rd International Conference on Machine Learning*, Seoul, South Korea. PMLR 306, 2026. Copyright 2026 by the author(s).

## 1. Introduction

As Large Language Models (LLMs) show remarkable performance across tasks (Brown et al., 2020; Achiam et al., 2023; Li et al., 2024). Aligning Large Language Models (LLMs) with complex human values is a cornerstone of modern AI safety and utility (Weidinger et al., 2021; Ji et al., 2023; Xu et al., 2025). The efficacy of this alignment process, typically driven by Reinforcement Learning from Human Feedback (RLHF) (Christiano et al., 2017), fundamentally depends on the reliability of the reward model used to proxy human judgments. Preference learning algorithms typically employ pairwise comparisons to capture human judgments (Ibarz et al., 2018; Ziegler et al., 2019). While the Bradley-Terry (BT) model (Bradley & Terry, 1952) serves as the standard (Bai et al., 2022; Rafailov et al., 2023), its reliance on scalar rewards enforces a strict transitivity assumption (i.e., $A \succ B \land B \succ C \Rightarrow A \succ C$). However, real-world human preferences are inherently heterogeneous and often exhibit intransitive, cyclic patterns (e.g., Rock-Paper-Scissors dynamics) (Tversky, 1969; Savage Jr, 1994; Gehrlein, 2006; Munos et al., 2024), which scalar models fundamentally fail to capture.

Early approaches like PairRM/PairPM (Jiang et al., 2023; Dong et al., 2024) predict preferences directly from concatenated inputs, theoretically capturing intransitive dynamics. However, their inference complexity ($O(K^2)$) limits applicability in scalable alignment scenarios. Recognizing the non-transitive nature of preferences, the General Preference Model (GPM) (Zhang et al., 2025c) maps responses to latent embeddings, modeling these dynamics via skew-symmetric bilinear forms with linear complexity.

While GPM effectively models cyclic dynamics, it remains underexplored whether its form can simultaneously preserve *transitive* global hierarchies (e.g., safety, helpfulness) while handling *cyclic* nuances. In this paper, we identify a critical theoretical gap: GPM cannot guarantee the representation of dominant solutions in arbitrary cyclic contexts due to the entangling of these properties. This insight motivates we revisit preference modeling from a game-theoretic perspective, treating the pairwise preference relation as a symmetric zero-sum game (Balduzzi et al., 2019). By leveraging the theoretical insight that any such game can be de-

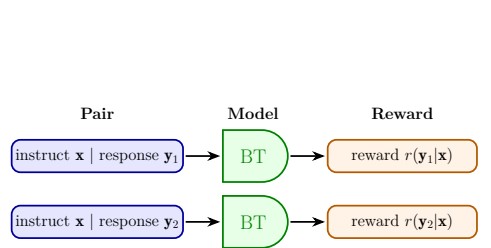

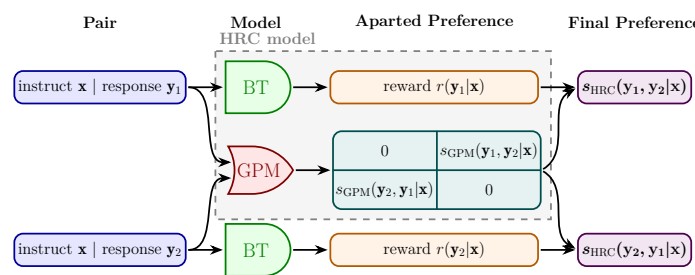

*Figure 1.* **Comparison of the Bradley-Terry (BT) model and the proposed Hybrid Reward-Cyclic (HRC) model.** (a) The BT model maps each instruction-response pair to a scalar reward, assuming transitive preferences. (b) The HRC model explicitly decomposes preferences into a transitive scalar component (via BT) and a cyclic vector component (via GPM), combining them to produce the final preference signal $s_{\text{HRC}}$.

composed, we propose the **Hybrid Reward-Cyclic (HRC)** model. Unlike prior approaches like GPM (Zhang et al., 2025c), HRC explicitly disentangles human preferences into two orthogonal components: a *transitive scalar component* that captures consistent global rankings, and a *cyclic vector component* that models intransitive local dynamics. This explicit decomposition not only enhances interpretability but also introduces an explicit inductive bias for modeling dominant hierarchies, allowing HRC to achieve superior performance while maintaining linear inference complexity $O((2d + 1)K)$, where $d \geq 1$ is a hyperparameter.

Furthermore, to effectively align LLMs with HRC, we focus on the **game-theoretic alignment** paradigm (e.g., SPPO (Wu et al., 2025b), INPO (Zhang et al., 2025b)). We prioritize this class of algorithms because they optimize directly against preference probabilities, thereby preserving the non-transitive modeling capabilities. However, existing frameworks in this category typically rely on a static preference oracle, which limits their ability to navigate complex preference landscapes. Treating preference fixed ignores the multi-dimensional structure of preferences revealed by HRC. To address this, we introduce **Dynamic Self-Play Preference Optimization (DSPPO)**, a generalized framework for time-varying preference games. Inspired by the success of curriculum learning (Bengio et al., 2009; Hacohen & Weinshall, 2019), we leverage DSPPO to orchestrate an optimization trajectory that transitions from a robust transitive backbone to refined cyclic nuances. This strategy stabilizes training by establishing a global quality baseline before introducing complex local dynamics, ensuring convergence to the Nash Equilibrium of the full preference game.

Our main contributions are summarized as follows:

- We propose the **HRC model**, which theoretically unifies BT model and GPM. By explicitly decomposing preferences into transitive and cyclic components, HRC model resolves the structural limitations of BT model and GPM.

- We introduce **DSPPO**, an alignment algorithm that leverages the HRC structure to schedule the complexity of the preference signal, enabling more robust convergence in complex landscapes.

- Extensive experiments on synthetic data constructed from UltraFeedback (Cui et al., 2024), RewardBench2 (Malik et al., 2025), AlpacaEval 2.0 (Dubois et al., 2024), Arena-Hard-v0.1 (Li et al., 2025), and MT-Bench (Zheng et al., 2023) demonstrate that our framework consistently outperforms existing baselines in both preference modeling accuracy and downstream generation quality.

## 2. Related Work

### 2.1. Preference Modeling in RLHF

In the standard RLHF framework (Christiano et al., 2017), human preferences are predominantly modeled using the Bradley-Terry (BT) model (Bradley & Terry, 1952), which typically employs a learnable reward function $r(\mathbf{y}|\mathbf{x})$ to assign a scalar score to each response. To explore alternative preference models within the RLHF framework, several approaches have been proposed, such as the Plackett-Luce model for K-wise comparisons (Zhu et al., 2023), Energy-Based Model for modeling preference distributions (Hong et al., 2025), and multi-dimensional reward systems like ArmoRM (Wang et al., 2024). However, despite these structural variations, these methods fundamentally rely on scalar scores or linear aggregations for final decision-making. Consequently, they implicitly retain the transitivity assumption (i.e., if $A \succ B$ and $B \succ C$, then $A \succ C$), thereby restricting their capacity to explicitly represent the complex, intransitive (cyclic) patterns often present in heterogeneous human feedback (Tversky, 1969; Munos et al., 2024).

Recent works have focused on explicitly modeling intransitive preferences in RLHF. While early pairwise approaches (e.g., PairPM/PairRM) (Jiang et al., 2023; Dong et al., 2024)

can capture cyclic patterns, they suffer from prohibitive quadratic inference costs ($O(K^2)$) when ranking $K$ candidate responses. To address this scalability bottleneck, GPM (Zhang et al., 2025c) employs a low-rank real skew-symmetric formulation that offers three key advantages: (1) controllable expressiveness via the latent dimension hyperparameter $d$; (2) superior efficiency with a linear inference complexity of $O(2dK)$; and (3) the inherent capability to model cyclic dynamics for any $d \geq 1$.

## 2.2. Preference-Based Reinforcement Learning from Human Feedback

Traditional RLHF (Christiano et al., 2017) typically follows a two-stage paradigm: learning a scalar reward model $r(\mathbf{y}|\mathbf{x})$ as a proxy for human preferences, followed by policy optimization via algorithms like PPO (Schulman et al., 2017). Recently, some works have proposed preference-based alignment methods that optimize the policy using preference probabilities as signals. A representative approach is IPO (Azar et al., 2024), which formulates a direct optimization objective based on these probabilities.

More recently, a rigorous game-theoretic perspective has gained prominence, framing LLM alignment as solving a multi-player zero-sum game to find the Nash Equilibrium. NLHF (Munos et al., 2024) first introduced this formulation to the field. Following this direction, a line of works—including SPO (Swamy et al., 2024), DNO (Rosset et al., 2024), SPPO (Wu et al., 2025b), INPO (Zhang et al., 2025b), GPO (Zhang et al., 2025c), EGPO (Zhou et al., 2025), ONPO (Zhang et al., 2025a), and MNPO (Wu et al., 2025a) have focused on optimizing policies directly from preference signals, thereby mitigating the limitations of scalar reward functions.

# 3. Preliminaries

We consider the standard language generation setting. A generative language model $\pi$ maps a prompt $\mathbf{x} = (x_1, x_2, \dots) \sim \mathcal{X}$ to a probability distribution over responses, from which we can sample candidate sequences $\mathbf{y}$ and $\mathbf{y}'$. The probability of generating any response $\mathbf{y} = (y_1, \dots, y_T)$ is defined via the autoregressive factorization $\pi(\mathbf{y}|\mathbf{x}) = \prod_{t=1}^{T} \pi(y_t|\mathbf{x}, \mathbf{y}_{<t})$, where $\mathbf{y}_{<t}$ denotes the partial sequence.

## 3.1. Preference Models

We assume a preference oracle provides binary feedback $o(\mathbf{y} \succ \mathbf{y}'|\mathbf{x}) \in \{0, 1\}$ on response pairs. The probability that $\mathbf{y}$ is preferred to $\mathbf{y}'$ is denoted as the expectation $\mathbb{P}(\mathbf{y} \succ \mathbf{y}'|\mathbf{x}) = \mathbb{E}[o(\mathbf{y} \succ \mathbf{y}'|\mathbf{x})]$.

In the standard RLHF framework (Christiano et al., 2017),

preferences are typically modeled using the Bradley-Terry (BT) model (Bradley & Terry, 1952). This model assumes a latent reward function $r(\mathbf{y}|\mathbf{x})$, defining the preference probability as $\mathbb{P}_{BT}(\mathbf{y} \succ \mathbf{y}'|\mathbf{x}) = \sigma(r(\mathbf{y}|\mathbf{x}) - r(\mathbf{y}'|\mathbf{x}))$, where $\sigma$ is the logistic function. However, the BT model can not capture intransitive preferences effectively (Bertrand et al., 2023). The nature of BT implies transitivity, where $r(\mathbf{y}) > r(\mathbf{y}')$ and $r(\mathbf{y}') > r(\mathbf{y}'')$ strictly enforce $r(\mathbf{y}) > r(\mathbf{y}'')$, prohibiting the representation of cyclic preferences.

To capture intransitive dynamics, recent works (Zhang et al., 2025c) propose the General Preference Model (GPM). Unlike the scalar formulation in BT, GPM maps the pair to latent vectors $\mathbf{v}(\mathbf{y}|\mathbf{x}) \in \mathbb{R}^{2d}$ and defines the preference score via a skew-symmetric operator:

$$s_{\text{GPM}}(\mathbf{y}, \mathbf{y}'|\mathbf{x}) = \mathbf{v}(\mathbf{y}|\mathbf{x})^\top \mathbf{W} \mathbf{v}(\mathbf{y}'|\mathbf{x}), \qquad (1)$$

where $\mathbf{W} \in \mathbb{R}^{2d \times 2d}$ is a real skew-symmetric matrix ($\mathbf{W}^\top = -\mathbf{W}$). The final probability is $\mathbb{P}_{\text{GPM}} = \sigma(s_{\text{GPM}})$. Due to the skew-symmetry, $s(\mathbf{y}, \mathbf{y}') = -s(\mathbf{y}', \mathbf{y})$, enabling the model to naturally represent cyclic preference structures in dimensions $2d \geq 2$.

## 3.2. Preference-based Reinforcement Learning from Human Feedback

Traditional alignment approaches, such as PPO (Schulman et al., 2017), rely on the BT model (Bradley & Terry, 1952) to maximize an expected scalar reward subject to a KL-divergence constraint. While effective, this scalar formulation implicitly assumes preference transitivity. Recently some works construct algorithms directly based on preference signals. Given a preference oracle $\mathbb{P}(\mathbf{y} \succ \mathbf{y}'|\mathbf{x})$, the alignment process is viewed as finding a policy $\pi_\theta$ that outperforms a competitor $\pi'$. For instance, the IPO algorithm (Azar et al., 2024) optimizes against a fixed competitor $\mu$:

$$\max_\theta \mathbb{E}_{\mathbf{x} \sim \mathcal{X}} \Big[ \mathbb{E}_{\mathbf{y} \sim \pi_\theta, \mathbf{y}' \sim \mu}[\mathbb{P}(\mathbf{y} \succ \mathbf{y}'|\mathbf{x})] \\ - \beta \mathbb{D}_{\text{KL}}(\pi_\theta(\cdot|\mathbf{x}) || \pi_{\text{ref}}(\cdot|\mathbf{x})) \Big], \qquad (2)$$

where $\mu$ typically represents naother fixed policy and $\beta$ controls the KL-divergence penalty $\mathbb{D}_{\text{KL}}$.

In game-theoretic approaches such as NLHF (Munos et al., 2024), SPPO (Wu et al., 2025b), DNO (Rosset et al., 2024), INPO (Zhang et al., 2025b), and GPO (Zhang et al., 2025c), the problem is formulated as a two-player constant-sum game. To simplify the formulation, NLHF (Munos et al., 2024) defines the preference of $\pi_\theta$ over $\pi'$ as the expectation of the preference signal over their generated responses:

$$\mathbb{P}(\pi_\theta \succ \pi'|\mathbf{x}) = \mathbb{E}_{\mathbf{y} \sim \pi_\theta(\cdot|\mathbf{x}), \mathbf{y}' \sim \pi'(\cdot|\mathbf{x})}[\mathbb{P}(\mathbf{y} \succ \mathbf{y}'|\mathbf{x})]. \quad (3)$$

With this notation, the alignment objective can be succinctly expressed as finding the Nash Equilibrium of the game,

which corresponds to the solution of the maximin problem:

$$\max_{\theta} \min_{\pi'} \mathbb{E}_{\mathbf{x} \sim \mathcal{X}} [\mathbb{P}(\pi_\theta \succ \pi' | \mathbf{x})]. \qquad (4)$$

# 4. Hybrid Reward-Cyclic Model

## 4.1. Modeling Human Preferences via Game-Theoretic Decomposition

In this section, we provide a theoretical grounding for our proposed architecture. We first formalize human preference modeling as a Symmetric Zero-Sum Functional-Form Game (FFG) and then leverage the game-theoretic decomposition analysis (Balduzzi et al., 2019) to justify the separation of transitive and cyclic components.

Consider a prompt $\mathbf{x}$ and a pair of responses $(\mathbf{y}_i, \mathbf{y}_j)$. We define the preference score in Equation (5).

**Definition 4.1** (Preference Score). Given a preference probability $\mathbb{P}(\mathbf{y}_i \succ \mathbf{y}_j | \mathbf{x})$, the preference score function $s(\mathbf{y}_i, \mathbf{y}_j | \mathbf{x})$ is defined as:

$$s(\mathbf{y}_i, \mathbf{y}_j | \mathbf{x}) \triangleq \log \frac{\mathbb{P}(\mathbf{y}_i \succ \mathbf{y}_j | \mathbf{x})}{1 - \mathbb{P}(\mathbf{y}_i \succ \mathbf{y}_j | \mathbf{x})}. \qquad (5)$$

This implies $\mathbb{P}(\mathbf{y}_i \succ \mathbf{y}_j | \mathbf{x}) = \sigma(s(\mathbf{y}_i, \mathbf{y}_j | \mathbf{x}))$.

To analyze the topological structure of these scores, we introduce two structural assumptions that map the discrete preference problem into a continuous vector space suitable for functional analysis.

**Assumption 4.2** (Skew-Symmetry). The preference relation is strictly skew-symmetric. For any pair $(\mathbf{y}_i, \mathbf{y}_j)$, $s(\mathbf{y}_i, \mathbf{y}_j | \mathbf{x}) = -s(\mathbf{y}_j, \mathbf{y}_i | \mathbf{x})$, which implies $s(\mathbf{y}_i, \mathbf{y}_i | \mathbf{x}) = 0$.

**Assumption 4.3** (Embedding Mapping). There exists a mapping function $g : \mathcal{Y} \times \mathcal{X} \to \mathbb{R}^{2d}$ such that each response $\mathbf{y}$ is represented by a vector $\mathbf{v} = g(\mathbf{y} | \mathbf{x})$.

**Definition 4.4** (Preference Function). The preference score can be expressed as a functional over these embeddings: $\phi_{\mathbf{x}}(\mathbf{v}_i, \mathbf{v}_j) \triangleq s(\mathbf{y}_i, \mathbf{y}_j | \mathbf{x})$.

Under these two assumptions, the function $\phi_{\mathbf{x}} : \mathbb{R}^{2d} \times \mathbb{R}^{2d} \to \mathbb{R}$ constitutes a Symmetric Zero-Sum Functional-Form Game (FFG) (Balduzzi et al., 2019).

We now invoke a fundamental result from previous work (Balduzzi et al., 2019). Theorem 4.5 provides the theoretical legitimacy for our dual-model approach.

**Theorem 4.5.** *Under two assumptions, any preference function $\phi(\mathbf{v}, \mathbf{w})$ can be uniquely decomposed into the sum of a transitive component $\phi_T$ and a cyclic component $\phi_C$: $\phi(\mathbf{v}, \mathbf{w}) = \phi_T(\mathbf{v}, \mathbf{w}) + \phi_C(\mathbf{v}, \mathbf{w})$, where:*

- *Transitive Component: There exists a potential function $f : \mathbb{R}^{2d} \to \mathbb{R}$ such that $\phi_T(\mathbf{v}, \mathbf{w}) = f(\mathbf{v}) - f(\mathbf{w})$. This component represents the hierarchical ranking capability of the game.*
- *Cyclic Component: The component $\phi_C$ satisfies $\int \phi_C(\mathbf{v}, \mathbf{w}) d\mathbf{w} = 0$ for all $\mathbf{v}$ (assuming a uniform measure over the embedding space). This component captures pure rotational dynamics where no global winner exists (e.g., Rock-Paper-Scissors dynamics).*

We include the proof of Theorem 4.5 in Section B.1.

Based on Theorem 4.5, we now instantiate the transitive component $\phi_T$ and the cyclic component $\phi_C$.

**Theorem 4.6.** *The decomposition admits a structural instantiation where $\phi_T$ corresponds to the **Bradley-Terry (BT)** model, and $\phi_C$ corresponds to the **General Preference Model (GPM)**, provided that the embeddings satisfy the zero-mean condition $\mathbb{E}[\mathbf{v}] = \mathbf{0}$.*

Based on Theorem 4.6 (see proof in Section B.2), we propose the **Hybrid Reward-Cyclic (HRC)** model, which explicitly instantiates this decomposition:

$$s_{\text{HRC}}(\mathbf{y}_i, \mathbf{y}_j | \mathbf{x}) = \underbrace{(r(\mathbf{y}_i) - r(\mathbf{y}_j))}_{\text{Transitive}} + \underbrace{(\mathbf{v}_i^\top \mathbf{W} \mathbf{v}_j)}_{\text{Cyclic}}, \qquad (6)$$

where $\mathbf{v}_i, \mathbf{v}_j$ correspond to the embeddings of $\mathbf{y}_i, \mathbf{y}_j$, and $\mathbf{W}$ is a real skew-symmetric matrix. We enforce the unit-norm constraint $\|\mathbf{v}\|_2 = 1$. Under the assumption that the embeddings are isotropically distributed on the hypersphere, the expectation vanishes ($\mathbb{E}[\mathbf{v}] = \mathbf{0}$), thereby satisfying the zero-mean condition required for the cyclic component. Crucially, HRC maintains a linear inference complexity of $O((2d + 1)K)$, matching the computational efficiency of both the BT model and GPM.

## 4.2. Implementation

We parameterize the decomposed components using a shared language model with three distinct projection heads. Let $\mathbf{h}_{\mathbf{y} | \mathbf{x}}$ denote the final hidden state of response $\mathbf{y}$ given prompt $\mathbf{x}$.

**Transitive and Cyclic Heads.** To capture the transitive component $\phi_T$, we employ a scalar reward head. To ensure numerical stability and mitigate potential gradient issues caused by unbounded scores, we apply a value clipping operation to constrain the reward within a fixed range $[-\delta, \delta]$:

$$r_\phi(\mathbf{y} | \mathbf{x}) = \text{clip}(\mathbf{w}_r^\top \mathbf{h}_{\mathbf{y} | \mathbf{x}}, -\delta, \delta).$$

For the cyclic component $\phi_C$, we utilize an embedding head that projects the hidden state into a latent space $\mathbb{R}^{2d}$. Following the GPM formulation (Zhang et al., 2025c), we

enforce a unit-norm constraint to ensure the zero-integral property:

$$\mathbf{v}(\mathbf{y}|\mathbf{x}) = \frac{\mathbf{W}_c \mathbf{h}_{\mathbf{y}|\mathbf{x}}}{\|\mathbf{W}_c \mathbf{h}_{\mathbf{y}|\mathbf{x}}\|_2}, \text{ where } \mathbf{v} \in \mathbb{R}^{2d}, \|\mathbf{v}\|_2 = 1.$$

**Context-Aware Gating.** To model context-dependent cyclic intensity, a gating mechanism computes a scaling matrix $\mathbf{D}(\mathbf{x})$. We ensure the generated diagonal elements are non-negative to adhere to the standard definition of spectral decomposition:

$$\lambda(\mathbf{x}) = g_\theta(\mathbf{h}_{\mathbf{x}}), \mathbf{D}(\mathbf{x}) = \text{diag}(\lambda(\mathbf{x})) \otimes \mathbf{I}_2, \text{ with } \lambda(\mathbf{x}) \geq 0.$$

**Training Objective.** The final preference score integrates both components with weighting hyperparameters $C_1, C_2$. We optimize the model end-to-end using the binary cross-entropy loss:

$$\mathcal{L}(\theta) = -\mathbb{E}_{(\mathbf{x}, \mathbf{y}_w, \mathbf{y}_l) \sim \mathcal{D}} \Big[ \log \sigma \Big( C_1 \big( r_\phi(\mathbf{y}_w) - \\ r_\phi(\mathbf{y}_l) \big) + C_2 \big( \mathbf{v}_w^\top \mathbf{D}(\mathbf{x}) \mathbf{R}^\succ \mathbf{D}(\mathbf{x}) \mathbf{v}_l \big) \Big) \Big]. \quad (7)$$

### 4.3. The Relationship between HRC and its Sub-components

GPM (Zhang et al., 2025c) has demonstrated capability in modeling cyclic preferences (e.g., Rock-Paper-Scissors). However, its structure introduces inherent limitations regarding transitive dominance. We analyze this through the concept of a "Dominant Candidate".

**Limitations of GPM on Transitivity.** To rigorously analyze the expressiveness of GPM, consider a set of responses forming a cycle $C = \{A_1, \ldots, A_n\}$ (where $A_1 \succ \cdots \succ A_n \succ A_1$) and a dominant candidate $A^*$ that strictly defeats all elements in the cycle (i.e., $A^* \succ A_i, \forall A_i \in C$). The following theorem characterizes the capability of GPM to model such simultaneous structures.

**Theorem 4.7.** *For the General Preference Model (GPM) with embedding dimension $2d$:*

- *Existence Condition: A configuration of a cycle $C$ and a candidate $A^*$ satisfying the dominance condition (i.e., $A^* \succ A_i, \forall A_i \in C$) can be validly represented in the embedding space $\mathbb{R}^{2d}$ if and only if $d > 1$.*
- *Lack of Arbitrariness: For any fixed finite $2d$, GPM cannot guarantee such representation for arbitrary cycles. Specifically, there exists a cycle $C$ such that no embedding in $\mathbb{R}^{2d}$ can simultaneously preserve the cyclic structure and represent a candidate $A^*$ satisfying the dominance condition.*

Theorem 4.7 (see proof in Section B.3) indicates that attempting to model global hierarchies (transitivity) implic-

itly through high-dimensional rotation is structurally brittle—complex local cycles can essentially "crowd out" the geometric capacity required to represent a dominant solution. In contrast, HRC resolves this by explicitly decoupling the two dynamics. By assigning a sufficiently large scalar reward $r(A^*)$ within its independent BT component, HRC can mathematically override the bounded residuals of the cyclic component. Consequently, HRC satisfies *Dominant Arbitrariness*, ensuring the preservation of the dominant candidate regardless of the complexity of the underlying cycle modeled by GPM.

While HRC is conceptually a hybrid ensemble of BT and GPM, it can be theoretically unified under the GPM framework. Since the BT model corresponds to a rank-1 GPM with embedding $\mathbf{v}_{\mathbf{y}} = [r(\mathbf{y}|\mathbf{x}), c]^\top$ ($c \neq 0$), HRC is theoretically equivalent to a constrained GPM of dimension $2d+1$. Its embedding takes the form $\mathbf{v}_{\mathbf{y}}^{HRC} = [r(\mathbf{y}|\mathbf{x}), c, \mathbf{v}_{\mathbf{y}}']^\top$, where the explicit transitive term $r(\mathbf{y}|\mathbf{x})$ acts as a robust shortcut for modeling dominant structures. This explains HRC's superiority over standard GPMs (dim=$2d$) which struggle to resolve such hierarchies purely.

## 5. Dynamic Self-Play Preference Optimization

### 5.1. Balancing Optimization Trajectories via Time-Varying Games

As established in Section 4.1, HRC model explicitly decomposes complex preferences into orthogonal transitive and cyclic dimensions. Alignment thus becomes a multi-dimensional optimization problem. To fully leverage this expressiveness, we require an algorithm capable of navigating this complex landscape.

Standard game-theoretic alignment methods (e.g., SPPO (Wu et al., 2025b), INPO (Zhang et al., 2025b)) typically seek the Nash Equilibrium of a static game defined by a fixed preference model. However, anchoring to a static target limits the model's ability to fully explore and exploit the unique alignment benefits inherent in the different dimensions of the generalized preference structure. To address this, we propose **Dynamic Self-Play Preference Optimization (DSPPO)**, which reformulates the alignment process as a **Time-Varying Game**. Instead of anchoring to a fixed target, DSPPO optimizes against a dynamic sequence of payoff functions that evolves over time.

### 5.2. Theoretical Framework and Algorithm

We begin by revisiting the update rule of Self-Play Policy Optimization (SPPO) (Wu et al., 2025b). In SPPO, the policy is updated to approximate the Nash Equilibrium of a constant-sum game defined by a fixed preference model $\mathbb{P}$. In addition, we define the winning probability of one response $\mathbf{y}$ against a distribution of responses $\pi$ as $\mathbb{P}(\mathbf{y} \succ$

$\pi|\mathbf{x}) = \mathbb{E}_{\mathbf{y}' \sim \pi(\cdot|\mathbf{x})}[\mathbb{P}(\mathbf{y} \succ \mathbf{y}'|\mathbf{x})]$. The iterative update is given by:

$$\pi_{t+1} = \arg\min_{\pi} \mathbb{E}_{\mathbf{x} \sim \mathcal{X}, \mathbf{y} \sim \pi_t(\cdot|\mathbf{x})}[(\log \frac{\pi(\mathbf{y}|\mathbf{x})}{\pi_t(\mathbf{y}|\mathbf{x})} - \eta(\widehat{\mathbb{P}}(\mathbf{y} \succ \pi_t|\mathbf{x}) - \frac{1}{2}))^2], \quad (8)$$

where $\widehat{\mathbb{P}}(\mathbf{y} \succ \pi_t|\mathbf{x}) = \frac{1}{K} \sum_{k=1}^{K} \mathbb{P}(\mathbf{y} \succ \mathbf{y}_k|\mathbf{x})$ is empirically estimated via $K$ samples. Effectively, this performs a multiplicative weight update:

$$\pi_{t+1}(\mathbf{y}|\mathbf{x}) \propto \pi_t(\mathbf{y}|\mathbf{x}) \exp(\eta \mathbb{P}(\mathbf{y} \succ \pi_t|\mathbf{x})). \quad (9)$$

However, the assumption of a static $\mathbb{P}$ throughout training restricts the flexibility of the optimization trajectory. Real-world preferences often exhibit nuanced multi-dimensional dynamics, which are effectively disentangled by advanced preference models like HRC model. Constraining the alignment process to a fixed proxy limits the algorithm's ability to exploit these rich structures for dynamic guidance. To address this, we propose **Dynamic Self-Play Preference Optimization (DSPPO)**, which allows the preference model $\mathbb{P}$ to evolve over time.

**Problem Formulation.** Let $s_\infty(\mathbf{y}, \mathbf{y}') = \log \frac{\mathbb{P}(\mathbf{y} \succ \mathbf{y}')}{1 - \mathbb{P}(\mathbf{y} \succ \mathbf{y}')}$ denote perference score of the real preference model $\mathbb{P}(\mathbf{y} \succ \mathbf{y}')$. We introduce a sequence of time-varying preference scores $\{s_t(\mathbf{y}, \mathbf{y}')\}_{t=1}^{T}$ and their corresponding probabilities $\mathbb{P}_t(\mathbf{y} \succ \mathbf{y}') = \sigma(s_t(\mathbf{y}, \mathbf{y}'))$. We make the following standard assumptions:

> **Assumption 5.1** (Boundedness). Both the dynamic and true scores are bounded, i.e., $s_t, s_\infty \in [-\rho, \rho]$ for some constant $\rho > 0$.

> **Assumption 5.2** (Convergence). The dynamic scores converge to the true scores effectively. Specifically, let $\epsilon_t = \max_{y,y'} |s_\infty(y, y') - s_t(y, y')|$. We assume the average error decays at a rate of $O(1/\sqrt{T})$, i.e., $\frac{1}{T} \sum_{t=1}^{T} \epsilon_t = O(1/\sqrt{T})$. A typical schedule satisfying this is $\epsilon_t = O(1/\sqrt{t})$.

At each iteration $t$, DSPPO optimizes the policy via the following learning objective:

$$\pi_{t+1} = \arg\min_{\pi} \mathbb{E}_{\mathbf{x} \sim \mathcal{X}, \mathbf{y} \sim \pi_t(\cdot|\mathbf{x})}[(\log \frac{\pi(\mathbf{y}|\mathbf{x})}{\pi_t(\mathbf{y}|\mathbf{x})} - \eta(\widehat{\mathbb{P}}_t(\mathbf{y} \succ \pi_t|\mathbf{x}) - \frac{1}{2}))^2], \quad (10)$$

where $\widehat{\mathbb{P}}_t$ is computed using the current preference model $\mathbb{P}_t$ like SPPO does.

> **Theorem 5.3.** *Assume the optimization oracle is realizable. Let $\pi_t$ be the policy obtained at step $t$ and $\bar{\pi}_T = \frac{1}{T} \sum_{t=1}^{T} \pi_t$ be the mixture policy. By setting the learning rate $\eta = \Theta(1/\sqrt{T})$, we have that:*
>
> $$\max_{\pi} \mathbb{P}(\pi \succ \bar{\pi}_T) - \min_{\pi} \mathbb{P}(\pi \prec \bar{\pi}_T) = O(\frac{1}{\sqrt{T}}). \quad (11)$$

Theorem 5.3 (see proof in Section B.4) characterizes the $O(1/\sqrt{T})$ convergence rate of the DSPPO's average policy toward the Nash equilibrium in terms of the duality gap, demonstrating robustness to time-varying preference signals provided the preference model converges sufficiently fast.

### 5.3. Implementation of DSPPO with HRC model

We now instantiate the general DSPPO framework using our proposed HRC model. As showed in Section 4.1, HRC model decomposes the preference score into transitive and cyclic components: $s_\infty = s_T + s_C$. To construct a dynamic schedule, we introduce a hyperparameter $\lambda$ and define the time-varying score $s_t$ as:

$$s_t = \left(1 + \frac{\lambda}{\sqrt{t}}\right) s_T + \left(1 - \frac{\lambda}{\sqrt{t}}\right) s_C. \quad (12)$$

In this work, we primarily adopt $\lambda > 0$ to regulate the convergence trajectory; we investigate impact of the $\lambda$ in DSPPO including the regime of $\lambda < 0$ and alternative schedule forms in Section C.4. As detailed in Section 4.2, we enforce explicit constraints on the model outputs: the transitive score $s_T$ is clipped to the range $[-\delta, \delta]$, while the boundedness of the cyclic score $s_C$ is inherently guaranteed by the GPM (Zhang et al., 2025c). Consequently, $s_t$, being a linear combination of strictly bounded terms, is guaranteed to remain within a compact range $[-\rho, \rho]$. Second, regarding convergence: The approximation error is given by $|s_\infty - s_t| = |\frac{\lambda}{\sqrt{t}} s_C - \frac{\lambda}{\sqrt{t}} s_T| \le \frac{|\lambda|}{\sqrt{t}}(|s_C| + |s_T|)$. Letting $C = \max(|s_C| + |s_T|)$, we have $\epsilon_t \le \frac{|\lambda|C}{\sqrt{t}}$. Consequently, $\frac{1}{T} \sum \epsilon_t \approx \frac{1}{T} \int |\lambda| C t^{-1/2} dt = O(1/\sqrt{T})$, fulfilling the convergence condition of Theorem 5.3.

## 6. Experiments

### 6.1. Modeling Cyclic Preference

To directly validate the "dominant + cycle" setting motivating HRC, we construct synthetic preference datasets based on UltraFeedback (Cui et al., 2024), following prior work (Zhang et al., 2025c). Each prompt contains four candidate responses with multi-dimensional annotations. We derive (i) cyclic preferences among three responses, and (ii) dominant + cycle settings by introducing an additional response that consistently outperforms the others.

We train GPM and HRC under identical configurations and

observe a two-stage learning behavior: models first identify the dominant candidate (accuracy improves from ∼50% to ∼75%), and then learn cyclic preferences (75% to 100%). HRC (dim=2+1, 4+1) finishes the first stage faster and achieves higher final accuracy than GPM (dim=4), while low-dimensional GPM (dim=2) fails to capture cyclic structure. These results provide direct empirical support that HRC more effectively models mixed transitive–cyclic preference structures.

Details of the dataset construction and experimental settings are provided in Section C.1.

### 6.2. Preference Modeling Capability

**Experimental Setup and Evaluation.** We evaluate preference modeling capabilities using Gemma-2B-it (Team et al., 2024a) and Llama-3.1-8B-Instruct (Grattafiori et al., 2024), trained on the Skywork-Reward-Preference-80K-v0.2 dataset (Liu et al., 2024). We benchmark our HRC model against BT model(Bradley & Terry, 1952) and GPM (Zhang et al., 2025c) on RewardBench 2 (Malik et al., 2025). This benchmark is selected for its rigorous Best-of-N ($N = 4$) evaluation format and its unique **Ties** domain, which effectively tests the model's robustness in distinguishing valid answers from incorrect ones without overfitting to noise among equivalently valid options. For detailed experimental setups, refer to Section C.3. To facilitate a rigorous comparison with the results reported in GPM (Zhang et al., 2025c), we independently conducted evaluations and report the results on the RewardBench (Lambert et al., 2025) in Section C.5.

**Results and Analysis.** The results are presented in Table 1. Across both model scales, HRC model consistently surpassing **both** the Bradley-Terry (BT) and General Preference Model (GPM) baselines. On the **Gemma-2B-it**, HRC achieves an average accuracy of **57.63%** (dim=$2 + 1$). This performance exceeds the strongest baseline in this category (56.40%) by **1.23%**, and significantly outperforms the scalar BT model (55.93%). Notably, HRC dominates in complex reasoning tasks: in the **Factuality** domain, it surpasses both baselines (BT: 45.68%, GPM: 47.16%) to reach **47.58%**, demonstrating superior capability in detecting subtle hallucinations. For the larger **Llama-3.1-8B-Instruct**, simple scalar models perform surprisingly well, with BT (70.10%) slightly edging out GPM (70.08%). However, HRC (dim=$2 + 1$) breaks this ceiling, achieving an average score of **70.95%**, which represents a robust improvement over the best-performing baseline (BT) by **0.85%**. Crucially, in the **Ties** subset—which tests the ability to handle equivalent correct answers without forcing arbitrary rankings—HRC demonstrates consistent superiority. It achieves **74.51%**, outperforming both the strict transitivity of BT (73.53%) and GPM (73.53%). This confirms that HRC's

structure successfully combines the transitive rewards with the cyclic embeddings, allowing it to resolve preference landscapes that neither BT nor GPM can master alone.

**Ablation Studies.** Our main results confirm that the explicit decoupling of transitive and cyclic components is essential for performance. To further validate the specific architectural choices that enable this effective decomposition, we conducted comprehensive ablation studies against the following variants:

- **w/o Context-Aware Gating**: We replace the input-dependent gating matrix $D(\mathbf{x})$ with a learnable static scalar parameter $\lambda$. The preference score becomes $s = r(\mathbf{y}_w) - r(\mathbf{y}_l) + \lambda \mathbf{v}_w^\top \mathbf{W} \mathbf{v}_l$. This variant tests whether the model requires dynamic adjustment of the cyclic component's weight based on the input prompt.
- **w/o Reward Clipping**: We remove the clipping operation on the scalar reward $r(\mathbf{y})$, allowing it to take values in $(-\infty, \infty)$.
- **w/o Unit Norm**: We remove the $\|\mathbf{v}\|_2 = 1$ constraint on the cyclic embeddings, allowing the vector magnitude to vary freely during training.

We find that removing the Context-Aware Gating mechanism or stability constraints leads to performance degradation, particularly in Focus. The results of ablation studies are showed in Table 2.

**Empirical Verification of Theoretical Assumptions.** For the zero-mean condition mentioned in Theorem 4.6, however, embeddings deviate from $\mathbb{E}[\mathbf{v}] = 0$. For example, HRC (dim=2+1) trained on the Skywork-Reward dataset with Gemma-2B-it, we obtain $\mathbb{E}[\mathbf{v}] \approx (0.13, -0.51)$. This suggests that the GPM component may indeed capture some transitive signal. Nevertheless, as shown in Table 1, HRC remains more effective than GPM in capturing complex preference structures despite such deviations.

### 6.3. Downstream Performance on Aligning Language Models with Human Preferences

We further investigate the effectiveness of our framework in language model alignment. Our evaluation focuses on two key aspects: (1) the quality of preference signals provided by our HRC model compared to BT model (Bradley & Terry, 1952) and GPM (Zhang et al., 2025c), and (2) the alignment efficiency of DSPPO algorithm compared to standard Self-Play Policy Optimization (SPPO).

**Experimental Setup and Baselines.** For the implementation of SPPO and DSPPO, we sampled responses using prompts derived from the UltraFeedback dataset (Cui et al., 2024). To rigorously evaluate the contribution of each component, we design the following comparative settings: (1) Impact of Preference Models. We fix the alignment algorithm to SPPO (Wu et al., 2025b) and vary the source of preference

*Table 1.* Comparison of preference modeling capabilities on **RewardBench 2** (Malik et al., 2025). We evaluate the baselines (BT, GPM) and our HRC model using **Gemma-2B-it** and **Llama-3.1-8B-Instruct** across six domains: Factuality, Precise Instruction Following (IF), Math, Safety, Focus, and Ties. Note that the BT and HRC models can be viewed as special cases of GPM with specific embedding constraints. The highest scores are highlighted in **bold**.

| BASE MODEL & METHOD | FACTUALITY | PRECISE IF | MATH | SAFETY | FOCUS | TIES | AVERAGE |
|---|---|---|---|---|---|---|---|
| GEMMA-2B-IT + BT (DIM=1) | 45.68 | 32.50 | 62.30 | 80.67 | 77.17 | 37.25 | 55.93 |
| GEMMA-2B-IT + GPM (DIM=2) | 47.16 | 33.75 | 62.84 | 78.00 | 71.92 | 38.24 | 55.32 |
| GEMMA-2B-IT + GPM (DIM=4) | 43.16 | **36.25** | **64.48** | 81.11 | 76.16 | 37.25 | 56.40 |
| GEMMA-2B-IT + HRC (DIM=2 + 1) | **47.58** | 35.63 | 61.75 | 82.00 | **79.60** | **39.22** | **57.63 (+1.23)** |
| GEMMA-2B-IT + HRC (DIM=4 + 1) | 45.89 | 33.75 | 62.30 | **83.78** | 77.78 | **39.22** | 57.12 |
| LLAMA-3.1-8B + BT (DIM=1) | 64.63 | 34.38 | **64.48** | 92.67 | 90.91 | 73.53 | 70.10 |
| LLAMA-3.1-8B + GPM (DIM=2) | 66.53 | 33.75 | 62.30 | 92.22 | 94.14 | 71.57 | 70.08 |
| LLAMA-3.1-8B + GPM (DIM=4) | 67.58 | 33.12 | 57.92 | 92.22 | 93.74 | 73.53 | 69.69 |
| LLAMA-3.1-8B + HRC (DIM=2 + 1) | **68.42** | **35.00** | 60.11 | **92.89** | 94.75 | **74.51** | **70.95 (+0.85)** |
| LLAMA-3.1-8B + HRC (DIM=4 + 1) | 68.00 | 32.50 | 61.20 | 92.67 | **95.15** | **74.51** | 70.67 |

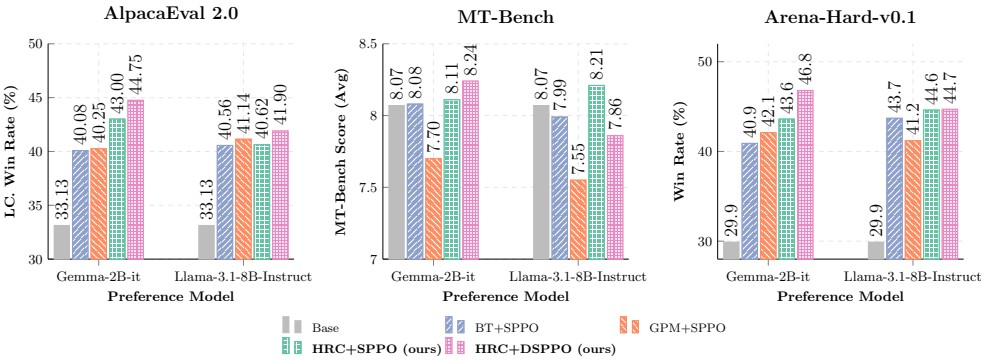

*Figure 2.* **Iteration 3 Alignment Performance across Three Benchmarks.** We compare BT+SPPO, GPM+SPPO, HRC+SPPO, and HRC+DSPPO at Iteration 3 on AlpacaEval 2.0 (LC. Win Rate), MT-Bench (Average Score), and Arena-Hard-v0.1 (Win Rate). Each panel reports results using both Gemma-2B-it and Llama-3.1-8B-Instruct as preference models.

signals: BT model (Bradley & Terry, 1952), GPM (Zhang et al., 2025c) and our HRC model. (2) Impact of Alignment Algorithms. We compare the SPPO against DSPPO (We employ the implementation of DSPPO with HRC model as described in Section 5.3, and set $\lambda = 1$), using the HRC model as the underlying preference oracle for both. The combination of HRC + DSPPO represents our full method. For detailed experimental setups, refer to Section C.3.

**Evaluation.** Our preliminary experiments and previous work (Zhang et al., 2025c) indicate that GPM-based alignment tends to yield longer responses. To comprehensively evaluate our framework, we employ three complementary benchmarks: (1) **AlpacaEval 2.0** (Dubois et al., 2024) focuses on length-controlled win rates to mitigate verbosity bias; (2) **MT-Bench** (Zheng et al., 2023) assesses multi-turn conversation and instruction-following capabilities; and (3) **Arena-Hard-v0.1** (Li et al., 2025) provides a rigorous assessment on challenging prompts requiring complex reasoning. All benchmarks use GPT-family models as evaluators for consistent quality assessment: GPT-4o-mini for

AlpacaEval 2.0 (selected for efficiency over GPT-4-Turbo while maintaining high agreement with human preferences), GPT-4 for MT-Bench, and GPT-4-Turbo for Arena-Hard-v0.1. We acknowledge that relying on GPT-based evaluation may introduce potential judge bias. To address this concern, we conduct additional validation analyses, including GPT vs. human agreement studies, inter-annotator agreement measurements, and cross-evaluation using multiple GPT variants. These analyses are detailed in Section C.7, demonstrating that our GPT-based evaluation pipeline produces reliable and robust results.

**Results and Analysis.** The comparative results in Figure 2 reveal four key insights. First, the **HRC model** establishes a stronger foundation than BT and GPM, achieving consistently higher baselines across all metrics. Second, **DSPPO** significantly enhances alignment efficiency, particularly on Gemma-2B-it, where it pushes the LC. WR to a peak of **44.75%** (+1.75% over static HRC) on AlpacaEval 2.0 and maintains robustness on MT-Bench (8.29 vs. 7.70 for GPM), effectively preventing the optimization collapse

*Table 2.* Ablation study on **RewardBench 2**. We evaluate the impact of removing key components (Context-Aware Gating, Reward Clipping, Unit Norm) across two models (**Gemma-2B-it**, **Llama-3.1-8B-Instruct**) and two dimension settings ($2 + 1$, $4 + 1$). The **Full Model** serves as the baseline, and $\Delta$ denotes the drop in Average Accuracy compared to the Full Model. The highest scores in each group are in **bold**.

| MODEL SETTING | FACTUALITY | PRECISE IF | MATH | SAFETY | FOCUS | TIES | AVERAGE | $\Delta$ |
|---|---|---|---|---|---|---|---|---|
| *Base Model: Gemma-2B-it* | | | | | | | | |
| *Dimension Setting:* $2 + 1$ | | | | | | | | |
| HRC (FULL) | 47.58 | **35.63** | 61.75 | 82.00 | **79.60** | 39.22 | **57.63** | - |
| W/O CONTEXT GATING | 45.89 | 35.00 | 61.75 | **83.11** | 74.95 | 38.24 | 56.49 | ↓ 1.14 |
| W/O REWARD CLIPPING | **47.79** | **35.63** | 61.75 | 81.33 | 75.96 | **40.20** | 57.11 | ↓ 0.52 |
| W/O UNIT NORM | 46.95 | 34.38 | **63.39** | 81.11 | 76.97 | 38.24 | 56.84 | ↓ 0.79 |
| *Dimension Setting:* $4 + 1$ | | | | | | | | |
| HRC (FULL) | 45.89 | 33.75 | 62.30 | **83.78** | 77.78 | **39.22** | 57.12 | - |
| W/O CONTEXT GATING | **46.53** | 35.00 | 61.75 | 82.00 | 78.99 | 36.27 | 56.76 | ↓ 0.36 |
| W/O REWARD CLIPPING | 42.74 | **36.88** | 63.93 | 82.00 | **81.62** | 36.27 | **57.24** | ↑ 0.12 |
| W/O UNIT NORM | 45.68 | 35.63 | 61.20 | 81.56 | 77.37 | **39.22** | 56.78 | ↓ 0.34 |
| *Base Model: Llama-3.1-8B-Instruct* | | | | | | | | |
| *Dimension Setting:* $2 + 1$ | | | | | | | | |
| HRC (FULL) | **68.42** | 35.00 | 60.11 | **92.89** | **94.75** | 74.51 | **70.95** | - |
| W/O CONTEXT GATING | 66.53 | 35.63 | **60.66** | 92.22 | **94.75** | 73.53 | 70.55 | ↓ 0.40 |
| W/O REWARD CLIPPING | 67.16 | **36.25** | 60.11 | 92.67 | 93.94 | 73.53 | 70.61 | ↓ 0.34 |
| W/O UNIT NORM | 68.00 | 32.50 | 57.92 | 92.44 | 93.54 | **75.49** | 69.98 | ↓ 0.97 |
| *Dimension Setting:* $4 + 1$ | | | | | | | | |
| HRC (FULL) | 68.00 | 32.50 | 61.20 | 92.67 | **95.15** | 74.51 | 70.67 | - |
| W/O CONTEXT GATING | 66.74 | **35.63** | 62.30 | **93.33** | 94.75 | 73.53 | 71.04 | ↑ 0.37 |
| W/O REWARD CLIPPING | 67.79 | 33.75 | 59.02 | 92.67 | **95.15** | 74.51 | 70.48 | ↓ 0.19 |
| W/O UNIT NORM | **68.42** | 35.00 | 61.75 | 92.89 | 94.95 | 74.51 | **71.25** | ↑ 0.58 |

observed in baselines. Third, as shown in Figure 2, the evaluation on Arena-Hard-v0.1 further validates the robustness of our framework. On Gemma-2B-it, HRC+DSPPO achieves a remarkable win rate of 46.8% in the final iteration, significantly outperforming both BT+SPPO (40.9%) and GPM+SPPO (42.1%). Notably, our method achieves a 3.2% improvement over the best baseline, demonstrating superior capability in handling challenging prompts that require complex reasoning. On Llama-3.1-8B-Instruct, HRC+DSPPO peaks at 45.5% in iteration 2 (shown in Table 8), showcasing consistent performance gains across different model scales. Finally, the results on Llama-3.1-8B-Instruct in iteration 2, where static HRC+SPPO marginally outperforms DSPPO on MT-Bench (8.42 vs. 8.14), suggest that alignment efficacy is sensitive to the hyperparameter $\lambda$. This indicates that while DSPPO promotes necessary exploration, larger models with stronger intrinsic capabilities may require a more conservative balance between the cyclic and transitive components to maintain stability in multi-turn reasoning. For additional evaluation metrics, refer to Section C.6.

**Scalability to Stronger Backbones.** To further assess the scalability of our framework, we conduct additional experiments applying the same preference models (trained on Gemma-2B-it) to post-train a larger backbone, Gemma-2-9B-it (Team et al., 2024b), and evaluate on AlpacaEval 2.0.

HRC+DSPPO achieves a peak length-controlled win-rate of **52.20%**, substantially outperforming BT+SPPO (48.79%) and the base model (38.38%). Detailed setup and results are provided in Section C.8.

## 7. Conclusion

In this paper, we revisited the mathematical foundations of preference modeling in RLHF, demonstrating that complex human preferences can be explicitly decomposed into two components: transitivity and cyclicity. To bridge this gap, we introduced the **Hybrid Reward-Cyclic (HRC)** model, a game-theoretic framework that explicitly decomposes preferences into transitive and cyclic components. Theoretical analysis confirms that HRC model overcomes the structural shortcomings of preference models such as BT and GPM. Complementing this, we propose **Dynamic Self-Play Preference Optimization (DSPPO)**, a novel alignment algorithm that treats policy optimization as a time-varying game. By incorporating the decomposed preference signals from HRC model to dynamically modulate the interplay between transitive guidance and cyclic exploration, DSPPO achieves a superior convergence trajectory. Extensive evaluations demonstrate that our framework consistently outperforms static baselines, leading to significant performance improvements in downstream tasks.

## Impact Statement

This paper presents work to advance the fundamental pursuit of aligning Artificial Intelligence with the rich complexity of human values. Faithfully representing such intricate preference structures is essential for developing AI systems that are not only robust and reliable but also intrinsically capable of respecting the heterogeneous nature of human judgment. We hope that advancements in this direction contribute to enabling future technologies to more safely and responsively navigate the diverse ethical and cultural landscapes of real-world deployment, potentially fostering greater trust in automated decision-making.

## Acknowledgements

This work was supported in part by National Key R&D Program of China (2024YFE0215300), National Natural Science Foundation of China (62476070), Shenzhen Science and Technology Program (JCYJ20241202123503005, GXWD20231128103232001, ZDSYS20230626091203008, KQTD20240729102154066), Department of Science and Technology of Guangdong (2024A1515011540), National Key R&D Program of China (SQ2024YFE0200592) and Suzhou Science and Technology Program (SYG2025072).

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

# Appendix

## A. Illustration of Our Motivation

To intuitively illustrate the complex duality of human preferences, we draw an analogy from the mechanics of the classic board game *L'Attaque* (or *Stratego*). The game dynamics perfectly encapsulate the tension between global quality (transitivity) and local stylistic trade-offs (cyclicity).

**1. Transitivity.** The fundamental structure of the game is built upon a strict, globally consistent chain of command.

- **Universal Dominance:** The ranking system (Marshal=10 . . . Scout=2) establishes a *Total Order*. A Marshal doesn't just defeat a General; by the property of transitivity, the Marshal is guaranteed to dominate *every* piece ranked lower than itself (Colonels, Captains, etc.).
- **Consistency:** If Piece $A$ defeats Piece $B$, and Piece $B$ defeats Piece $C$, logical consistency dictates that $A$ must defeat $C$. There is no ambiguity.

**2. Cyclicity.** However, strictly hierarchical systems cannot model complex interactions. The game introduces a singularity which breaks the total order and creates a loop: Mine.

- **The Intransitive Loop:** Consider the interaction between three specific units:
    1. **Marshal (10) $\succ$ Miner (3):** The Marshal wins easily due to superior rank (Quality).
    2. **Miner (3) $\succ$ Mine:** The Miner possesses a specific skill to defuse the Mine.
    3. **Mine $\succ$ Marshal (10):** The Mine destroys the Marshal upon contact.
- **Mathematical Impossibility for Scalar Scores:** This creates a closed cycle: *Marshal $\succ$ Miner $\succ$ Mine $\succ$ Marshal*. It is mathematically impossible to assign scalar values $v$ to these pieces such that $v(\text{Marshal}) > v(\text{Miner}) > v(\text{Mine}) > v(\text{Marshal})$.

**3. Hypothesis in Alignment.** Drawing from this analogy, we hypothesize that human preferences in language model alignment exhibit a similar duality. Preferences are inherently multi-dimensional, often adhering to a transitive hierarchy reflecting general quality (e.g., safety and truthfulness). However, the complexity of human values implies that specific response attributes may excel in particular dimensions—such as strict brevity or stylistic constraints—thereby forming intransitive loops that defy global ranking. This hypothesis is substantiated by foundational game-theoretic studies, which have formally decomposed interactions into transitive strength and cyclic "spinning top" geometries (Balduzzi et al., 2018; 2019; Czarnecki et al., 2020). Within the context of LLM alignment, (Zhang et al., 2025c) recently highlighted the prevalence of intransitive preferences and introduced the General Preference Model (GPM) to address them. Building upon these insights, our work adopts the explicit transitive-cyclic decomposition strategy to develop the HRC model, specifically designed to harmonize the hierarchy of response quality with cyclic stylistic variations.

## B. The Proofs of Theorems

### B.1. Proof of Theorem 4.5

Let $\Omega$ denote the strategy space of embeddings, and let $\mu$ be the uniform probability measure over $\Omega$. We assume the game $\phi(\mathbf{v}, \mathbf{w})$ is skew-symmetric, i.e., $\phi(\mathbf{v}, \mathbf{w}) = -\phi(\mathbf{w}, \mathbf{v})$.

**Transitive Component ($\phi_T$).** We define the potential function $f(\mathbf{v})$ as the expected payoff of strategy $\mathbf{v}$ against the uniform population:

$$f(\mathbf{v}) := \int_{\Omega} \phi(\mathbf{v}, \mathbf{w}) d\mu(\mathbf{w}) = \mathbb{E}_{\mathbf{w} \sim \mu}[\phi(\mathbf{v}, \mathbf{w})]. \tag{13}$$

Using this potential function, we define the transitive component as the difference in potentials:

$$\phi_T(\mathbf{v}, \mathbf{w}) := f(\mathbf{v}) - f(\mathbf{w}). \tag{14}$$

**Cyclic Component ($\phi_C$).** We define the cyclic component as the residual:

$$\phi_C(\mathbf{v}, \mathbf{w}) := \phi(\mathbf{v}, \mathbf{w}) - \phi_T(\mathbf{v}, \mathbf{w}). \tag{15}$$

We assume the game $\phi$ is skew-symmetric. Since $\phi_T$ is also skew-symmetric by construction ($f(\mathbf{v}) - f(\mathbf{w}) = -(f(\mathbf{w}) - f(\mathbf{v}))$), the residual $\phi_C$ must preserve skew-symmetry. Now, we verify the zero-marginal condition. Integrating $\phi_C$ with

respect to $\mathbf{w}$:

$$\int_\Omega \phi_C(\mathbf{v}, \mathbf{w}) d\mu(\mathbf{w}) = \int_\Omega \left( \phi(\mathbf{v}, \mathbf{w}) - (f(\mathbf{v}) - f(\mathbf{w})) \right) d\mu(\mathbf{w}) \tag{16}$$

$$= \underbrace{\int_\Omega \phi(\mathbf{v}, \mathbf{w}) d\mu(\mathbf{w})}_{=f(\mathbf{v})} - \int_\Omega f(\mathbf{v}) d\mu(\mathbf{w}) + \int_\Omega f(\mathbf{w}) d\mu(\mathbf{w}) \tag{17}$$

$$= f(\mathbf{v}) - f(\mathbf{v}) \cdot 1 + \mathbb{E}_\mathbf{w}[f(\mathbf{w})]. \tag{18}$$

To complete the proof, we must show that $\mathbb{E}_\mathbf{w}[f(\mathbf{w})] = 0$. Using the skew-symmetry of $\phi$:

$$\mathbb{E}_\mathbf{w}[f(\mathbf{w})] = \int_\Omega \int_\Omega \phi(\mathbf{w}, \mathbf{z}) d\mu(\mathbf{z}) d\mu(\mathbf{w}) = 0. \tag{19}$$

This double integral vanishes because $\phi(\mathbf{w}, \mathbf{z}) = -\phi(\mathbf{z}, \mathbf{w})$, and the integration domain is symmetric. Thus, we have:

$$\int_\Omega \phi_C(\mathbf{v}, \mathbf{w}) d\mu(\mathbf{w}) = 0. \tag{20}$$

This confirms that $\phi_C$ is a cycle game.

**Uniqueness.** Suppose there exists another decomposition $\phi = \phi'_T + \phi'_C$ where $\phi'_T(\mathbf{v}, \mathbf{w}) = g(\mathbf{v}) - g(\mathbf{w})$ for some potential $g$, and $\phi'_C$ satisfies the zero-marginal condition. We have:

$$\phi_T + \phi_C = \phi'_T + \phi'_C \implies \phi_T - \phi'_T = \phi'_C - \phi_C. \tag{21}$$

Integrating both sides with respect to $\mathbf{w}$ over $\Omega$:

$$\int_\Omega (\phi_T(\mathbf{v}, \mathbf{w}) - \phi'_T(\mathbf{v}, \mathbf{w})) d\mu(\mathbf{w}) = \int_\Omega (\phi'_C(\mathbf{v}, \mathbf{w}) - \phi_C(\mathbf{v}, \mathbf{w})) d\mu(\mathbf{w}). \tag{22}$$

By the definition of cyclic components, the RHS is $0 - 0 = 0$. Expanding the LHS:

$$\int_\Omega [(f(\mathbf{v}) - f(\mathbf{w})) - (g(\mathbf{v}) - g(\mathbf{w}))] d\mu(\mathbf{w}) = 0 \tag{23}$$

$$(f(\mathbf{v}) - g(\mathbf{v})) - \int_\Omega (f(\mathbf{w}) - g(\mathbf{w})) d\mu(\mathbf{w}) = 0. \tag{24}$$

Let $C = \int_\Omega (f(\mathbf{w}) - g(\mathbf{w})) d\mu(\mathbf{w})$ be a constant independent of $\mathbf{v}$. Then $f(\mathbf{v}) = g(\mathbf{v}) + C$. Substituting this back into the expression for $\phi_T$:

$$\phi_T(\mathbf{v}, \mathbf{w}) = f(\mathbf{v}) - f(\mathbf{w}) = (g(\mathbf{v}) + C) - (g(\mathbf{w}) + C) = g(\mathbf{v}) - g(\mathbf{w}) = \phi'_T(\mathbf{v}, \mathbf{w}). \tag{25}$$

Since $\phi_T = \phi'_T$, it immediately follows that $\phi_C = \phi - \phi_T = \phi - \phi'_T = \phi'_C$. Thus, the decomposition is unique.

### B.2. Proof of Theorem 4.6

According to the decomposition theory of Factorial Feature Games (FFGs) (Balduzzi et al., 2019), any FFG $\phi(\mathbf{y}, \mathbf{y}')$ can be uniquely decomposed into a transitive component $\phi_T$ and a cyclic component $\phi_C$, such that $\phi = \phi_T + \phi_C$. The defining properties are:

1. For any $\mathbf{y}, \mathbf{y}'$, the transitive component satisfies $\phi_T(\mathbf{y}, \mathbf{y}') = f(\mathbf{y}) - f(\mathbf{y}')$ for some potential function $f : \mathbb{R}^{2d} \to \mathbb{R}$.

2. The cyclic component has zero marginal utility against the population distribution, i.e., $\mathbb{E}_{\mathbf{y}'}[\phi_C(\mathbf{y}, \mathbf{y}')] = 0$ for all $\mathbf{y}$.

We now verify that using BT model and GPM explicitly instantiates this structure.

**BT model ($\phi_T$).** The theorem defines the transitive component as $\phi_T(\mathbf{y}_i, \mathbf{y}_j) = f(\mathbf{y}_i) - f(\mathbf{y}_j)$ for some potential function $f : \mathbb{R}^{2d} \to \mathbb{R}$. In the BC model, we have $s(\mathbf{y}_i, \mathbf{y}_j | \mathbf{x}) = r(\mathbf{y}_i | \mathbf{x}) - r(\mathbf{y}_j | \mathbf{x})$. By identifying the scalar reward function $r(\cdot)$

as the potential function $f(\cdot)$, the BT model strictly satisfies the definition of the transitive component. The scalar reward $r(\mathbf{y}|\mathbf{x})$ serves as the global "potential" or "rank" of the response $\mathbf{y}$ while the prompt is $\mathbf{x}$, capturing the hierarchical structure of preferences.

**GPM ($\phi_C$).** Let the second component be $\phi_C(\mathbf{y}, \mathbf{y}') = \mathbf{v}(\mathbf{y})^\top \mathbf{W} \mathbf{v}(\mathbf{y}')$, where $\mathbf{W}$ is real and skew-symmetric (Zhang et al., 2025c). First, $\phi_C$ is clearly skew-symmetric. Crucially, we investigate its transitive contribution by calculating its marginal expectation over the response distribution. Under the assumption that the embeddings are zero-mean centered (i.e., $\mathbb{E}_\mathbf{y}[\mathbf{v}(\mathbf{y})] = \mathbf{0}$):

$$\mathbb{E}_{\mathbf{y}'}[\phi_C(\mathbf{y}, \mathbf{y}')] = \mathbb{E}_{\mathbf{y}'}[\mathbf{v}(\mathbf{y})^\top \mathbf{W} \mathbf{v}(\mathbf{y}')] = \mathbf{v}(\mathbf{y})^\top \mathbf{W} \mathbb{E}_{\mathbf{y}'}[\mathbf{v}(\mathbf{y}')] = \mathbf{v}(\mathbf{y})^\top \mathbf{W} \cdot \mathbf{0} = 0. \tag{26}$$

Since the marginal expectation is zero for any $\mathbf{y}$, the GPM component contributes zero "strength" or "rank" to the model. This implies that $\phi_C$ is purely cyclic.

**Conclusion.** Since $\phi_T$ fully captures the transitive structure and $\phi_C$ (under the zero-mean condition) constitutes a purely cyclic structure orthogonal to $\phi_T$, the sum $s = \phi_T + \phi_C$ constitutes a valid structural instantiation of the FFG decomposition.

### B.3. Proof of Theorem 4.7

We begin by establishing the rigorous mathematical form of the General Preference Model (GPM) based on the spectral theory of real skew-symmetric matrices.

**Mathematical Formulation.** Let $\mathbf{M} \in \mathbb{R}^{n \times n}$ be the real skew-symmetric preference matrix where $M_{ij} = s(\mathbf{y}_i, \mathbf{y}_j)$. Since $\mathbf{M}$ is skew-symmetric ($\mathbf{M}^\top = -\mathbf{M}$), its rank is even, denoted as $2d$ (Horn & Johnson, 2012). By the spectral theorem for real skew-symmetric matrices, $\mathbf{M}$ can be decomposed into its real normal form:

$$\mathbf{M} = \sum_{l=1}^{d} \tau_l (\mathbf{q}_{2l-1} \mathbf{q}_{2l}^\top - \mathbf{q}_{2l} \mathbf{q}_{2l-1}^\top), \tag{27}$$

where $\tau_l > 0$ are the magnitudes of the imaginary parts of the eigenvalues, and $\{\mathbf{q}_1, \ldots, \mathbf{q}_{2d}\}$ are orthonormal vectors. Inspired by previous work (Chen & Joachims, 2016; Bertrand et al., 2023), we let $\mathbf{b}_l = \sqrt{\tau_l} \mathbf{q}_{2l-1}$ and $\mathbf{c}_l = \sqrt{\tau_l} \mathbf{q}_{2l}$, we can define matrices $\mathbf{B}, \mathbf{C} \in \mathbb{R}^{n \times d}$ such that their $l$-th columns correspond to these vectors. The preference matrix can then be factorized as:

$$\mathbf{M} = \mathbf{B}\mathbf{C}^\top - \mathbf{C}\mathbf{B}^\top. \tag{28}$$

For any pair of responses $(\mathbf{y}_i, \mathbf{y}_j)$, the preference score is the element $M_{ij}$. Let $\mathbf{b}_i, \mathbf{c}_i \in \mathbb{R}^d$ denote the $i$-th rows of $\mathbf{B}$ and $\mathbf{C}$ respectively. The score is:

$$s(\mathbf{y}_i, \mathbf{y}_j) = \mathbf{b}_i \mathbf{c}_j^\top - \mathbf{c}_i \mathbf{b}_j^\top = \sum_{l=1}^{d} (b_{il} c_{jl} - c_{il} b_{jl}). \tag{29}$$

Geometrically, each term $(b_{il} c_{jl} - c_{il} b_{jl})$ represents the determinant of the vectors $\mathbf{z}_{il} = [b_{il}, c_{il}]^\top$ and $\mathbf{z}_{jl} = [b_{jl}, c_{jl}]^\top$. Thus, Equation (29) can be rewritten in polar coordinates where $\mathbf{z}_{il}$ has magnitude $L_{il}$ and angle $\phi_{il}$:

$$s(\mathbf{y}_i, \mathbf{y}_j) = \sum_{l=1}^{d} L_{il} L_{jl} \sin(\phi_{jl} - \phi_{il}). \tag{30}$$

This formulation confirms that a rank-$2d$ GPM is equivalent to summing the cyclic interactions across $d$ independent 2D subspaces.

**Existence Condition.** Assume $d = 1$. The preference score simplifies to a single term: $s(\mathbf{y}_i, \mathbf{y}_j) = L_{i1} L_{j1} \sin(\phi_{j1} - \phi_{i1})$. For a set of candidates $C = \{A_1, \ldots, A_n\}$ to form a valid cycle (e.g., $A_1 \succ A_2 \succ \cdots \succ A_n \succ A_1$), their angles $\phi_{i1}$ must span the entire interval $[0, 2\pi)$ relative to the origin. However, for a dominant candidate $A^*$ to strictly defeat all $A_i \in C$, we require $s(\mathbf{y}_{A^*}, \mathbf{y}_{A_i}) > 0$ for all $i$. This implies:

$$\sin(\phi_{i1} - \phi_{A^*1}) > 0 \implies \phi_{i1} \in (\phi_{A^*1}, \phi_{A^*1} + \pi). \tag{31}$$

This requires all cycle candidates to be confined within a strictly open semi-circle, which geometrically contradicts the requirement of forming a complete cycle. Thus, $d = 1$ is insufficient.

We verify that if $d \geq 2$, GPM can represent a dominant candidate against a cycle. Consider $d = 2$. We construct a cycle $C = \{A_1, \ldots, A_n\}$ and a dominant candidate $A^*$ using the following embedding configuration:

1. For candidates $A_i \in C$, set $L_{i1} = 1$ and $\phi_{i1} = \frac{2\pi i}{n}$. For the dominant candidate $A^*$, set $L_{A^*1} = 0$.

2. For candidates $A_i \in C$, set identical vectors: $L_{i2} = 1, \phi_{i2} = 0$. For $A^*$, set $L_{A^*2} = 1, \phi_{A^*2} = -\frac{\pi}{2}$.

Now we evaluate the scores using Equation (30):

- Since $\phi_{i2} = \phi_{j2} = 0$, the contribution from the second subspace is $\sin(0) = 0$. The score is determined purely by the first subspace: $s(\mathbf{y}_{A_i}, \mathbf{y}_{A_j}) = 1 \cdot 1 \cdot \sin\left(\frac{2\pi(j-i)}{n}\right) + 0$, which correctly models the cyclic structure.
- The contribution from the first subspace is 0 (since $L_{A^*1} = 0$). The score is determined purely by the second subspace: $s(\mathbf{y}_{A^*}, \mathbf{y}_{A_i}) = 0 + L_{A^*2}L_{i2}\sin(\phi_{i2} - \phi_{A^*2}) = 1 \cdot 1 \cdot \sin\left(0 - \left(-\frac{\pi}{2}\right)\right) = 1 > 0$.

Since $s(\mathbf{y}_{A^*}, \mathbf{y}_{A_i})$ is strictly positive for all $i$, we find an example satisfied. Similarly, we can also prove that it holds true when $d > 2$.

**Lack of Arbitrariness.** We verify that for any fixed finite rank $2d$, there exists an example for which no embedding can satisfy conditions.

Consider $C = \{A_1, \ldots, A_n\}$ constructed such that the embedding vectors are identical and aligned across all $d$ subspaces. Let $\mathbf{z}_{i,m} \in \mathbb{R}^2$ denote the embedding vector of candidate $A_i$ in the $m$-th subspace. For the Hard Cycle, we define:

$$\forall m \in \{1, \ldots, d\}: \quad \mathbf{z}_{i,m} = \begin{bmatrix} \cos\theta_i \\ \sin\theta_i \end{bmatrix}, \quad \text{where } \theta_i = \frac{2\pi i}{n}. \tag{32}$$

The preference score between cycle members is $s(\mathbf{y}_{A_i}, \mathbf{y}_{A_j}) = \sum_{m=1}^{d} \det(\mathbf{z}_{i,m}, \mathbf{z}_{j,m}) = \sum_{m=1}^{d} \sin(\theta_j - \theta_i) = d\sin(\theta_j - \theta_i)$, which defines a valid cycle.

Now, assume there exists a dominant candidate $D$ parameterized by arbitrary embedding vectors $\mathbf{z}_{D,m} = [X_m, Y_m]^\top$ for each subspace $m = 1, \ldots, d$. According to the GPM preference score definition (sum of determinants across subspaces):

$$s(\mathbf{y}_D, \mathbf{y}_{A_i}) = \sum_{m=1}^{d} \det(\mathbf{z}_{D,m}, \mathbf{z}_{i,m}) = \sum_{m=1}^{d} (X_m \sin\theta_i - Y_m \cos\theta_i). \tag{33}$$

By linearity of the summation, we can group the coefficients for $\sin\theta_i$ and $\cos\theta_i$:

$$s(\mathbf{y}_D, \mathbf{y}_{A_i}) = \left(\sum_{m=1}^{d} X_m\right) \sin\theta_i - \left(\sum_{m=1}^{d} Y_m\right) \cos\theta_i. \tag{34}$$

Let $\mathcal{X} = \sum_{m=1}^{d} X_m$ and $\mathcal{Y} = -\sum_{m=1}^{d} Y_m$. The expression simplifies to a single harmonic wave form:

$$s(\mathbf{y}_D, \mathbf{y}_{A_i}) = \sqrt{\mathcal{X}^2 + \mathcal{Y}^2} \sin(\theta_i + \delta), \tag{35}$$

where $\delta$ is a constant. For $D$, we require the strict inequality $s(\mathbf{y}_D, \mathbf{y}_{A_i}) > 0$ to hold for all $i = 1, \ldots, n$. However, the function $f(\theta) = A\sin(\theta + \delta)$ has a mean of zero over the period $[0, 2\pi]$. Since the candidates $A_i$ have angles $\theta_i$ distributed uniformly over the circle, as $n$ increases, there must exist candidates located in the phase where $\sin(\theta_i + \delta) < 0$.

Therefore, it is mathematically impossible to find a set of embedding vectors $\{\mathbf{z}_{D,m}\}$ for $D$ that yields a positive score against all members of this cycle. This proves that GPM cannot guarantee the representation of dominant hierarchies in arbitrary cyclic candidates.

### B.4. Proof of Theorem 5.3

We assume the optimization oracle in Equation (10) is realizable. The iterative update rule of DSPPO is given by:

$$\pi_{t+1}(\mathbf{y}|\mathbf{x}) \propto \pi_t(\mathbf{y}|\mathbf{x}) \exp\left(\eta \mathbb{P}_t(\mathbf{y} \succ \pi_t | \mathbf{x})\right), \quad \text{for } t = 1, 2, \ldots, T. \tag{36}$$

**Step 1: Regret Bound for Time-Varying Preferences**

We first extend Theorem 1 from (Freund & Schapire, 1999). Since $\mathbb{P}_t$ acts as a valid preference oracle at each step $t$, we can invoke the regret bound for the multiplicative weights update algorithm. For any sequence of reference mixed policies $\mu_1, \mu_2, \ldots, \mu_T$ and the sequence of policies $\pi_1, \pi_2, \ldots, \pi_T$ generated by Equation (36), the following inequality holds (based on Lemma 2 in (Freund & Schapire, 1999)):

$$\sum_{t=1}^{T} \mathbb{P}_t(\pi_t \prec \mu_t) \leq \min_{\pi} \left[ \frac{\eta}{1 - e^{-\eta}} \sum_{t=1}^{T} \mathbb{P}_t(\pi \prec \mu_t) + \frac{\mathbb{D}_{\text{KL}}(\pi||\pi_0)}{1 - e^{-\eta}} \right]. \tag{37}$$

Setting $\mu_t = \pi_t$, and noting that $\mathbb{P}_t(\pi_t \prec \pi_t) = 1/2$ (a policy ties with itself), the LHS simplifies to $T/2$. Using the skew-symmetry property $\mathbb{P}(\pi \prec \mu) = 1 - \mathbb{P}(\pi \succ \mu)$, we have:

$$\frac{T}{2} \leq \min_{\pi} \left[ \frac{\eta T}{1 - e^{-\eta}} \frac{1}{T} \sum_{t=1}^{T} \mathbb{P}_t(\pi \prec \pi_t) + \frac{\mathbb{D}_{\text{KL}}(\pi||\pi_0)}{1 - e^{-\eta}} \right]. \tag{38}$$

Rearranging the terms and dividing by $T$, we obtain:

$$\frac{1 - e^{-\eta}}{2\eta} \leq \min_{\pi} \left[ \frac{1}{T} \sum_{t=1}^{T} \mathbb{P}_t(\pi \prec \pi_t) + \frac{\mathbb{D}_{\text{KL}}(\pi||\pi_0)}{\eta T} \right]. \tag{39}$$

Using the Taylor expansion $\frac{1 - e^{-\eta}}{2\eta} = \frac{1}{2} - \frac{\eta}{4} + O(\eta^2)$, and substituting $\mathbb{P}_t(\pi \prec \pi_t) = 1 - \mathbb{P}_t(\pi \succ \pi_t)$, we get:

$$\frac{1}{2} - \frac{\eta}{4} + O(\eta^2) \leq 1 - \max_{\pi} \left[ \frac{1}{T} \sum_{t=1}^{T} \mathbb{P}_t(\pi \succ \pi_t) \right] + \frac{\mathbb{D}_{\text{KL}}(\pi||\pi_0)}{\eta T}. \tag{40}$$

Rearranging to isolate the win rate:

$$\max_{\pi} \left[ \frac{1}{T} \sum_{t=1}^{T} \left( \mathbb{P}_t(\pi \succ \pi_t) - \frac{1}{2} \right) \right] \leq \frac{\eta}{4} + \frac{\mathbb{D}_{\text{KL}}(\pi||\pi_0)}{\eta T} + O(\eta^2). \tag{41}$$

Since $\pi_0$ is an autoregressive model fully supported on a finite vocabulary, $\| \log \pi_0(\cdot) \|_\infty$ is bounded. Thus, $\mathbb{D}_{\text{KL}}(\pi||\pi_0) \leq \| \log \pi_0(\cdot) \|_\infty$. By choosing the learning rate $\eta = \Theta(1/\sqrt{T})$, specifically $\eta = \frac{\| \log \pi_0(\cdot) \|_\infty}{\sqrt{T}}$, the RHS is bounded by $O(1/\sqrt{T})$. Thus:

$$\max_{\pi} \frac{1}{T} \sum_{t=1}^{T} \left[ \mathbb{P}_t(\pi \succ \pi_t) - \frac{1}{2} \right] = O\left( \frac{1}{\sqrt{T}} \right). \tag{42}$$

**Step 2: Bridging Time-Varying Preferences to True Preferences**

We define the optimality gap to the mixture policy $\bar{\pi}_T = \frac{1}{T} \sum_{t=1}^{T} \pi_t$ under the true preference model $\mathbb{P}$. We decompose the gap into approximation error (due to time-varying $s_t$) and optimization error:

$$\max_{\pi} \left[ \mathbb{P}(\pi \succ \bar{\pi}_T) - \frac{1}{2} \right] = \max_{\pi} \left[ \frac{1}{T} \sum_{t=1}^{T} \mathbb{P}(\pi \succ \pi_t) - \frac{1}{2} \right] \tag{43}$$

$$= \max_{\pi} \left[ \frac{1}{T} \sum_{t=1}^{T} (\mathbb{P}(\pi \succ \pi_t) - \mathbb{P}_t(\pi \succ \pi_t)) + \frac{1}{T} \sum_{t=1}^{T} \left( \mathbb{P}_t(\pi \succ \pi_t) - \frac{1}{2} \right) \right] \tag{44}$$

$$\leq \underbrace{\max_{\pi} \frac{1}{T} \sum_{t=1}^{T} (\mathbb{P}(\pi \succ \pi_t) - \mathbb{P}_t(\pi \succ \pi_t))}_{\text{Approximation Error}} + \underbrace{\max_{\pi} \frac{1}{T} \sum_{t=1}^{T} \left( \mathbb{P}_t(\pi \succ \pi_t) - \frac{1}{2} \right)}_{\text{Optimization Error}}. \tag{45}$$

The Optimization Error is $O(1/\sqrt{T})$ as shown in Equation (42). We now bound the Approximation Error. Recall that $\mathbb{P}(\mathbf{y} \succ \mathbf{y}') = \sigma(s_\infty(\mathbf{y}, \mathbf{y}'))$ and $\mathbb{P}_t(\mathbf{y} \succ \mathbf{y}') = \sigma(s_t(\mathbf{y}, \mathbf{y}'))$. The derivative of the sigmoid function satisfies

$\sigma'(x) = \sigma(x)(1 - \sigma(x)) \leq 1/4$ for all $x \in \mathbb{R}$. This implies that $\sigma(\cdot)$ is Lipschitz continuous with constant $L = 1/4$. Therefore:

$$|\mathbb{P}(\mathbf{y} \succ \mathbf{y}') - \mathbb{P}_t(\mathbf{y} \succ \mathbf{y}')| = |\sigma(s_\infty(\mathbf{y}, \mathbf{y}')) - \sigma(s_t(\mathbf{y}, \mathbf{y}'))| \leq \frac{1}{4}|s_\infty(\mathbf{y}, \mathbf{y}') - s_t(\mathbf{y}, \mathbf{y}')|. \tag{46}$$

Let $\epsilon_t = \max_{\mathbf{y}, \mathbf{y}'} |s_\infty(\mathbf{y}, \mathbf{y}') - s_t(\mathbf{y}, \mathbf{y}')|$. By the linearity of expectation, we have:

$$\mathbb{P}(\pi \succ \pi_t) - \mathbb{P}_t(\pi \succ \pi_t) = \mathbb{E}_{\mathbf{y} \sim \pi, \mathbf{y}' \sim \pi_t}[\sigma(s_\infty(\mathbf{y}, \mathbf{y}')) - \sigma(s_t(\mathbf{y}, \mathbf{y}'))] \leq \frac{1}{4}\epsilon_t. \tag{47}$$

Based on Theorem 5.2, we have $\frac{1}{T} \sum_{t=1}^{T} \epsilon_t = O(1/\sqrt{T})$. Thus, the Approximation Error is bounded by $O(1/\sqrt{T})$.

**Step 3: Final Bound**

Combining the bounds for both errors:

$$\max_\pi \left[\mathbb{P}(\pi \succ \bar{\pi}_T) - \frac{1}{2}\right] \leq O\left(\frac{1}{\sqrt{T}}\right) + O\left(\frac{1}{\sqrt{T}}\right) = O\left(\frac{1}{\sqrt{T}}\right). \tag{48}$$

Finally, the optimality gap is given by:

$$\text{Gap}(\bar{\pi}_T) = \max_\pi \mathbb{P}(\pi \succ \bar{\pi}_T) - \min_{\pi'} \mathbb{P}(\pi' \prec \bar{\pi}_T) \tag{49}$$

$$= \max_\pi \mathbb{P}(\pi \succ \bar{\pi}_T) - \min_{\pi'} (1 - \mathbb{P}(\pi' \succ \bar{\pi}_T)) \tag{50}$$

$$= \max_\pi \mathbb{P}(\pi \succ \bar{\pi}_T) - \left(1 - \max_{\pi'} \mathbb{P}(\pi' \succ \bar{\pi}_T)\right) \tag{51}$$

$$= 2\left(\max_\pi \mathbb{P}(\pi \succ \bar{\pi}_T) - \frac{1}{2}\right). \tag{52}$$

Substituting the bound derived above, we conclude:

$$\max_\pi \mathbb{P}(\pi \succ \bar{\pi}_T) - \min_{\pi'} \mathbb{P}(\pi' \prec \bar{\pi}_T) = O\left(\frac{1}{\sqrt{T}}\right). \tag{53}$$

This completes the proof.

# C. More on Experiments

## C.1. Cyclic Preference Setup

We construct synthetic "cycle" and "dominant + cycle" datasets based on UltraFeedback (Cui et al., 2024). Each prompt is associated with four candidate responses, each annotated along four dimensions (e.g., helpfulness, honesty, instruction-following, and truthfulness).

**Cyclic Dataset.** We select three dimensions and construct cyclic preferences by choosing three responses such that each response outperforms another on a different dimension:

$$A \succ B, \quad B \succ C, \quad C \succ A.$$

Pairwise preferences are defined according to the dimension on which the comparison is made.

**Dominant + Cycle Dataset.** We extend the above by selecting a fourth response that outperforms all others across the selected dimensions. This introduces three additional preference pairs, forming a mixed structure with both a dominant candidate and cyclic relations.

Using the constructed datasets, we train GPM and HRC under identical settings. We observe a consistent two-stage learning process:

- **Stage 1 ($50\% \rightarrow 75\%$):** models identify the dominant candidate, driven by the three dominant-related preference pairs.
- **Stage 2 ($75\% \rightarrow 100\%$):** models learn cyclic relations among the remaining candidates.

HRC finishes Stage 1 faster and achieves higher final accuracy, while low-dimensional GPM (dim=2) fails to capture cyclic structure.

## C.2. Preference Modeling Setup

Given our inclusion of the **RewardBench 2** (Malik et al., 2025), we independently trained all preference models (BT, GPM, and HRC) from scratch to ensure a rigorous and consistent evaluation pipeline. This approach ensures that all comparisons in both preference modeling (Section 6.2) and downstream alignment (Section 6.3) are based on models trained on the exact same data distribution, eliminating potential discrepancies arising from different pre-trained checkpoints.

**Datasets and Models.** All preference models are trained on the **Skywork-Reward-Preference-80K-v0.2** (Liu et al., 2024). We employ **Gemma-2B-it** (Team et al., 2024a) and **Llama-3.1-8B-Instruct** (Grattafiori et al., 2024) as the base models.

**Model Configurations and Dimensions.** To ensure a comprehensive evaluation, we adopt distinct configuration strategies for the two experimental phases:

*1. Preference Modeling (RewardBench 2 & RewardBench).* To assess the modeling capabilities and scaling properties of our proposed method, we evaluated models across varying latent dimensions as reported in Table 1:

- **Bradley-Terry (BT):** Fixed output dimension of 1.
- **General Preference Model (GPM):** Evaluated at latent dimensions $2d \in \{2, 4\}$.
- **Hybrid Reward-Cyclic (HRC):** Evaluated at dimensions $2d+1 \in \{2+1, 4+1\}$, corresponding to a cyclic component of dimension $2d$ augmented with a scalar reward.

*2. Downstream Alignment (AlpacaEval 2.0 & MT-Bench).* For the downstream policy optimization experiments (Section 6.3), we selected the high-capacity configurations to maximize performance differentiation and ensure a rigorous structural comparison. Specifically, we compare:

- **Baselines:** The standard BT model (dim=1) and the **GPM (dim=4)**.
- **Ours:** The **HRC (dim=4+1)**.

This selection allows for a strictly controlled comparison: HRC (dim=4+1) effectively augments the GPM (dim=4) baseline with an explicit transitive shortcut. By comparing these specific configurations, we isolate the architectural contribution of the explicit preference decomposition, verifying whether adding the transitive component improves alignment efficacy over the pure cyclic formulation of GPM.

## C.3. Implementation Details

Our experiments were implemented using the **PyTorch** framework (Paszke et al., 2019) and the **HuggingFace Transformers** library (Wolf et al., 2020). To ensure efficient distributed training, we leveraged **DeepSpeed** (Rajbhandari et al., 2020). Furthermore, our preference modeling codebase is developed based on the official GPM implementation (Zhang et al., 2025c), which is built upon the **OpenRLHF** framework (Hu et al., 2025).

For the preference modeling experiments (comparing BT, GPM, and HRC), we maintain a consistent training framework to ensure fair comparability.

**Unified Training Objective in Preference Models.** To establish a unified training paradigm for BT, GPM, and HRC, we formulate the learning objective as a generalized pairwise classification task. Let $s_\theta(\mathbf{x}, \mathbf{y}_w, \mathbf{y}_l)$ denote the **pairwise preference score** predicted by the model for a prompt $\mathbf{x}$ and a response pair $(\mathbf{y}_w, \mathbf{y}_l)$. The loss function is defined as:

$$\mathcal{L}(\theta) = -\mathbb{E}_{(\mathbf{x}, \mathbf{y}_w, \mathbf{y}_l) \sim \mathcal{D}} \left[ \log \sigma \left( \frac{s_\theta(\mathbf{x}, \mathbf{y}_w, \mathbf{y}_l)}{\tau} \right) \right], \tag{54}$$

where $\tau$ is the temperature hyperparameter (corresponding to general_preference_tau) that controls the sharpness of the preference distribution. This formulation generalizes all three models, where the distinction lies in the mathematical definition of the pairwise score $s_\theta$:

- **BT:** The pairwise score decomposes into the difference of scalar rewards: $s_\theta(\mathbf{x}, \mathbf{y}_w, \mathbf{y}_l) = r(\mathbf{x}, \mathbf{y}_w) - r(\mathbf{x}, \mathbf{y}_l)$.
- **General Preference Model (GPM):** The score is computed directly on the pair embeddings, modulated by a context-aware gate: $s_\theta(\mathbf{x}, \mathbf{y}_w, \mathbf{y}_l) = \phi(\mathbf{y}_w)^\top \mathbf{D}(\mathbf{x}) \mathbf{A} \mathbf{D}(\mathbf{x}) \phi(\mathbf{y}_l)$, where $\mathbf{A}$ is the skew-symmetric operator capturing intransitivity, and $\mathbf{D}(\mathbf{x})$ represents the diagonal gating matrix generated by the prompt head.
- **HRC:** The score is the sum of the transitive (scalar difference) and cyclic (GPM) components: $s_\theta(\mathbf{x}, \mathbf{y}_w, \mathbf{y}_l) = \underbrace{(r(\mathbf{x}, \mathbf{y}_w) - r(\mathbf{x}, \mathbf{y}_l))}_{\text{Transitive}} + \underbrace{s_{\text{GPM}}(\mathbf{x}, \mathbf{y}_w, \mathbf{y}_l)}_{\text{Cyclic}}$.

**Hyperparameters for Preference Model Training.** We specifically align our parameters with the settings reported in the GPM literature (Zhang et al., 2025c), utilizing a temperature $\tau = 0.1$. The detailed hyperparameters derived from our training scripts are listed in Table 3.

*Table 3.* **Hyperparameters for Preference Model Training.** These settings are applied consistently to BT, GPM, and HRC models.

| CONFIGURATION | VALUE |
|---|---|
| *Optimization Configuration* | |
| GPU TYPES | NVIDIA A800 80GB GPUs |
| GPUs | 4 |
| GLOBAL BATCH SIZE | 32 |
| MICRO BATCH SIZE | 1 |
| GRADIENT ACCUMULATION | 8 |
| LEARNING RATE | $2 \times 10^{-6}$ |
| MAX EPOCHS | 2 |
| OPTIMIZER | ADAMW |
| PRECISION | BF16 |
| MAX SEQUENCE LENGTH | 2048 |
| *Loss & Model Specifics* | |
| TEMPERATURE ($\tau$) | 0.1 |

**Iterative Self-Play Setup.** For the downstream alignment phase, we employ the iterative self-play framework. While the preference signal is provided by our pre-trained models, the policy optimization process itself involves specific generation and training configurations derived from the SPPO protocol (Wu et al., 2025b). To implement DSPPO, we dynamically modulated the output of the preference model during the computation of preference probabilities in the SPPO framework, adjusting the signal strength according to the iteration index. The alignment is conducted over $T = 3$ iterations. To estimate the preference landscape effectively, we generate $K = 5$ distinct responses for each prompt in the UltraFeedback dataset(Cui et al., 2024). We use a sampling temperature of $1.0$ to encourage diverse exploration of the policy's response space.

**Training Hyperparameters in Alignment.** The hyperparameters used for the policy updates are listed in Table 4. We strictly adhere to the settings provided in the official SPPO implementation to ensuring a fair evaluation of our proposed DSPPO scheduling strategy.

*Table 4.* **Hyperparameters for Alignment (SPPO & DSPPO).** Parameters are consistent across all iterations.

| CONFIGURATION | VALUE |
|---|---|
| *Optimization Configuration* | |
| GPU TYPES | NVIDIA A800 80GB GPUs |
| GPUs | 4 |
| OPTIMIZER | RMSPROP |
| LEARNING RATE | $5 \times 10^{-7}$ |
| GLOBAL BATCH SIZE | 64 |
| GRADIENT ACCUMULATION | 1 |
| MAX SEQUENCE LENGTH | 2048 |
| NUM EPOCHS PER ITERATION | 1 |
| LR SCHEDULER | CONSTANT |
| *Self-Play Dynamics* | |
| NUMBER OF ITERATIONS ($T$) | 3 |
| SAMPLED RESPONSES ($K$) | 5 |
| REGULARIZATION $\beta$ | 0.001 |
| SAMPLING TEMPERATURE | 1.0 |

## C.4. Impact of the $\lambda$ in DSPPO

In the DSPPO framework, the hyperparameter $\lambda$ controls the dynamic weighting between the transitive ($s_T$) and cyclic ($s_C$) components. Our proposed schedule uses $\lambda > 0$, which initializes the training with a stronger emphasis on the transitive component (global hierarchy) and progressively increases the weight of the cyclic component (local nuances). This design is motivated by the hypothesis of curriculum learning (Bengio et al., 2009; Hacohen & Weinshall, 2019): the model should

first establish a robust baseline of instruction following and safety (transitivity) before refining its behavior with complex, non-transitive stylistic preferences (cyclicity).

To validate this hypothesis, we conduct a comprehensive investigation across a broad range of $\lambda$ values. We systematically evaluate $\lambda \in \{-2, -1, -0.5, 0, 0.25, 0.5, 0.75, 1, 2\}$ using the Llama-3.1-8B-Instruct backbone. This analysis allows us to examine: (1) the effect of negative $\lambda$ values, which imply a reverse trajectory prioritizing cyclic dynamics early in training; (2) the impact of varying positive $\lambda$ values on the convergence trajectory; and (3) the behavior when $|\lambda| > 1$, which may cause the schedule coefficients to become negative.

**Results.** Table 5 presents the performance on AlpacaEval 2.0 (Iteration 3) across different $\lambda$ settings. Several key observations emerge from our analysis:

First, $\lambda > 0$ **generally leads to better performance**, with the proposed schedule achieving the peak Length-Controlled Win Rate (LC. WR) of **41.90%** at $\lambda = +1.0$. Among positive $\lambda$ values, smaller magnitudes (0.25, 0.5, 0.75) also perform well, with LC. WRs of 41.09%, 40.85%, and 41.54% respectively.

Second, $\lambda < 0$ **consistently underperforms**. Settings with $\lambda = -0.5, -1.0$, and $-2.0$ achieve LC. WRs of 40.21%, 38.66%, and 37.27% respectively. This validates our hypothesis that starting from stable, transitive preference signals and gradually incorporating cyclic components is beneficial for training, while the reverse trajectory struggles to establish a solid foundation.

Third, the **Static Baseline** ($\lambda = 0$) performs adequately with LC. WR of 40.62%, but plateaus and fails to reach the final alignment quality achieved by dynamic schedules.

Fourth, when $|\lambda| > 1$, the coefficients in the schedule $s_t = (1 + \frac{\lambda}{\sqrt{t}})s_T + (1 - \frac{\lambda}{\sqrt{t}})s_C$ may become negative for certain components. In this regime, the resulting signal lacks a clear semantic interpretation as a meaningful combination of multiple preference signals. Our experiments show that $\lambda = 2.0$ (LC. WR of 40.15%) underperforms compared to reasonable positive $\lambda$ values, further supporting the interpretation that extreme weightings compromise the schedule's effectiveness.

Additionally, we experimented with alternative schedule forms, such as replacing $\frac{\lambda}{\sqrt{t}}$ with $\frac{\lambda}{\sqrt[3]{t}}$ or using sinusoidal schedules (e.g., $s_t = (1 + \sin(\frac{\pi}{2t}))s_T + (1 - \sin(\frac{\pi}{2t}))s_C$). However, these variants did not exhibit stable or consistent improvements compared to the $1/\sqrt{t}$-based schedule.

Overall, these results suggest that DSPPO is not overly sensitive to precise $\lambda$ tuning within a reasonable range (approximately 0.25 to 1.0), and that the proposed $\lambda = +1.0$ schedule provides a stable and effective default choice.

*Table 5.* **Ablation study on scheduling parameter** $\lambda$ **(AlpacaEval 2.0, Iteration 3).** We evaluate a broad range of $\lambda$ values using the **Llama-3.1-8B-Instruct** preference model. Results are reported for the model obtained at the third iteration. **LC. WR**: Length-Controlled Win Rate (%); **WR**: Standard Win Rate; **Avg. Len**: Average response length. The best results are highlighted in **bold**. When $|\lambda| > 1$, coefficients in the schedule may become negative, potentially compromising semantic interpretation.

| $\lambda$ | LC. WR | WR | Avg. Len |
|---|---|---|---|
| BASE MODEL | 33.13 | 35.26 | 2106 |
| *Negative $\lambda$ (Inverse schedule)* | | | |
| $-2.0$ | 37.27 | 41.41 | 2170 |
| $-1.0$ | 38.66 | 43.09 | 2192 |
| $-0.5$ | 40.21 | 44.22 | 2179 |
| *Zero $\lambda$ (Static baseline)* | | | |
| 0.0 | 40.62 | 46.30 | 2245 |
| *Positive $\lambda$ (Proposed schedule)* | | | |
| 0.25 | 41.09 | 44.84 | 2183 |
| 0.50 | 40.85 | 42.98 | 2139 |
| 0.75 | 41.54 | 43.73 | 2139 |
| +1.0 | **41.90** | 44.79 | 2171 |
| *Extreme $\lambda > 1$* | | | |
| +2.0 | 40.15 | 44.30 | 2186 |

*Table 6.* Comparison of preference modeling capabilities on RewardBench. We evaluate the Bradley-Terry (BT) model, the General Preference Model (GPM), and HRC model using Gemma-2B-it and Llama-3.1-8B-Instruct. Note that the BT model and HRC model can be viewed as a special case of GPM with special embedding dimension. The highest scores are highlighted in **bold**.

| BASE MODEL&METHOD | CHAT | CHAT-HARD | SAFETY | REASONING | AVERAGE |
|---|---|---|---|---|---|
| GEMMA-2B-IT + BT(DIM=1) | 81.56 | 68.77 | 84.86 | 83.86 | 79.76 |
| GEMMA-2B-IT + GPM(DIM=2) | 83.24 | 70.39 | 85.00 | 85.32 | 80.99 |
| GEMMA-2B-IT + GPM(DIM=4) | 82.40 | 69.08 | 84.32 | 87.35 | 80.79 |
| GEMMA-2B-IT + HRC(DIM=2 + 1) | **84.64** | **71.05** | 85.68 | 87.42 | **82.20(+1.21)** |
| GEMMA-2B-IT + HRC(DIM=4 + 1) | 83.80 | 69.74 | **86.08** | **87.49** | 81.78 |
| LLAMA-3.1-8B-INSTRUCT + BT(DIM=1) | 89.11 | 84.86 | 92.97 | 94.90 | 90.46 |
| LLAMA-3.1-8B-INSTRUCT + GPM(DIM=2) | 92.74 | 85.30 | 91.76 | 94.76 | 91.14 |
| LLAMA-3.1-8B-INSTRUCT + GPM(DIM=4) | 92.46 | 83.91 | 92.16 | **95.81** | 91.08 |
| LLAMA-3.1-8B-INSTRUCT + HRC(DIM=2 + 1) | 93.58 | **85.96** | 92.70 | 95.11 | 91.84 |
| LLAMA-3.1-8B-INSTRUCT + HRC(DIM=4 + 1) | **94.13** | 85.53 | **93.11** | 95.18 | **91.99(+0.85)** |

## C.5. Additional Experiments on Rewardbench

To ensure a comprehensive comparison with established baselines in the literature, we extend our evaluation to the RewardBench (Lambert et al., 2025). Following the same experimental setup as our main analysis, we benchmark the HRC model against BT and GPM using the Gemma-2B-it and Llama-3.1-8B-Instruct trained on the Skywork-80K dataset.

**Results and Analysis.** The results are presented in Table 6. On RewardBench, using the Gemma-2B-it base model, HRC achieves an average score of 82.20% (dim=2 + 1), which is an improvement of 1.21% over the GPM baseline's best average score of 80.99%. Specifically, in the Chat task, HRC improves performance from 83.24% (GPM) to 84.64%, and in the Safety task, from 85.00% to 86.08%. For the Llama-3.1-8B-Instruct base model, HRC achieves an average score of 91.99% (dim=4 + 1), representing a 0.85% improvement over the GPM baseline's average score of 91.14%. In the Chat task, HRC improves from 92.74% (GPM) to 94.13%, demonstrating a significant lead over the baselines. These results indicate that HRC consistently outperforms both the BT and GPM baselines across various base models and tasks, particularly in the Chat and Safety categories which require capturing nuanced preferences and robust safety alignment. Note that the HRC model can be viewed as the combination of BT model and GPM.

*Table 7.* **AlpacaEval 2.0 evaluation results.** We compare alignment performance on Llama-3.1-8B-Instruct, using different methods: SPPO(BT, GPM, and HRC) and DSPPO(HRC). **LC. WR**: Length-Controlled Win Rate (%); **WR**: Standard Win Rate (%) against GPT-4-Turbo; **Avg. Len**: Average response length in tokens. The best scores within each iteration group are marked in **bold**.

| METHOD | ITER | PM: GEMMA-2B-IT | | | PM: LLAMA-3.1-8B-INSTRUCT | | |
|---|---|---|---|---|---|---|---|
| | | LC. WR | WR | AVG. LEN | LC. WR | WR | AVG. LEN |
| BASE | – | 33.13 | 35.26 | 2106 | 33.13 | 35.26 | 2106 |
| **BT+SPPO** | 1 | 37.35 | 39.00 | 2079 | 38.85 | 43.04 | 2160 |
| **GPM+SPPO** | 1 | 35.91 | 40.15 | 2167 | 38.94 | 43.80 | 2184 |
| **HRC+SPPO** | 1 | 39.72 | **44.04 (+0.48)** | 2224 | 38.58 | 42.87 | 2176 |
| **HRC+DSPPO** | 1 | **40.33 (+0.61)** | 43.56 | 2155 | **39.84 (+0.90)** | **43.93 (+0.13)** | 2170 |
| **BT+SPPO** | 2 | 38.75 | 40.27 | 2057 | 38.15 | 41.94 | 2161 |
| **GPM+SPPO** | 2 | 40.39 | 43.22 | 2109 | 39.08 | 42.86 | 2182 |
| **HRC+SPPO** | 2 | **40.84 (+0.38)** | **44.61 (+1.32)** | 2173 | 40.35 | **45.10 (+0.75)** | 2207 |
| **HRC+DSPPO** | 2 | 40.46 | 43.29 | 2143 | **40.74 (+0.39)** | 44.35 | 2209 |
| **BT+SPPO** | 3 | 40.08 | 41.78 | 2049 | 40.56 | 43.97 | 2165 |
| **GPM+SPPO** | 3 | 40.25 | 43.16 | 2168 | 41.14 | 45.36 | 2210 |
| **HRC+SPPO** | 3 | 43.00 | **46.99 (+0.06)** | 2191 | 40.62 | **46.30 (+0.94)** | 2245 |
| **HRC+DSPPO** | 3 | **44.75 (+1.75)** | 46.93 | 2111 | **41.90 (+0.76)** | 44.79 | 2171 |

## C.6. Detailed Analysis of Alignment Results

**Results and Analysis.** Complementing the summarized findings in the main text, we present the comprehensive evaluation breakdowns for AlpacaEval 2.0, Arena-Hard-v0.1, and MT-Bench in Table 7, Table 8, and Table 9, respectively. This analysis reveals three critical insights regarding the behavior of our proposed framework. First, the detailed metrics in

Table 7 demonstrate that HRC+DSPPO achieves its superior win rate (44.75%) through genuine capability improvements rather than length exploitation; unlike GPM baselines which exhibit signs of "reward hacking" by inflating response length (e.g., jumping to 2168 tokens), our method maintains concise outputs (2111 tokens), validating that the cyclic component enhances information density without encouraging verbosity. Second, the evaluation results on Arena-Hard-v0.1 in Table 8 further validate the robustness of our framework. On Gemma-2B-it, HRC+DSPPO achieves a remarkable score of 46.8% in the final iteration, significantly outperforming both BT+SPPO (40.9%) and GPM+SPPO (42.1%). Notably, our method achieves a 3.2% improvement over the best baseline, demonstrating superior capability in handling challenging prompts. On Llama-3.1-8B-Instruct, HRC+DSPPO peaks at 45.5% in iteration 2, showcasing consistent performance gains across different model scales. Third, the turn-level breakdown in Table 9 suggests that appropriate selection of the preference modeling framework can significantly improve alignment outcomes. For instance, using the preference model trained on the Gemma-2B-it, our HRC+DSPPO method successfully identifies a highly effective policy trajectory, achieving a peak score of 8.29. This performance notably surpasses the GPM baseline, which drops to 7.70 in the final iteration, indicating that our approach is capable of finding better solutions in complex, multi-turn scenarios.

*Table 8.* **Arena-Hard-v0.1 evaluation results.** We evaluate alignment performance on Arena-Hard-v0.1, a challenging benchmark that tests model capabilities on difficult prompts. We compare BT+SPPO, GPM+SPPO, HRC+SPPO, and HRC+DSPPO across three iterations using both **Gemma-2B-it** and **Llama-3.1-8B-Instruct** as preference models. **WR**: Standard Win Rate (%) against GPT-4-0314. The best scores within each iteration group are marked in **bold**.

| METHOD | ITER | PM: GEMMA-2B-IT WR | PM: LLAMA-3.1-8B-INSTRUCT WR |
|---|---|---|---|
| BASE (LLAMA3-8B) | – | 29.9 | 29.9 |
| BASE (GPT-4-0314) | – | 50.0 | 50.0 |
| BT+SPPO | 1 | 32.9 | **35.2 (+0.9)** |
| GPM+SPPO | 1 | 31.5 | 31.6 |
| HRC+SPPO | 1 | **39.3 (+2.6)** | 34.1 |
| HRC+DSPPO | 1 | 36.7 | 34.4 (+4.5) |
| BT+SPPO | 2 | 37.8 | 40.9 |
| GPM+SPPO | 2 | 39.7 | 34.4 |
| HRC+SPPO | 2 | **42.0 (+0.8)** | 41.4 |
| HRC+DSPPO | 2 | 41.2 | **45.5 (+4.1)** |
| BT+SPPO | 3 | 40.9 | 43.7 |
| GPM+SPPO | 3 | 42.1 | 41.2 |
| HRC+SPPO | 3 | 43.6 | 44.6 |
| HRC+DSPPO | 3 | **46.8 (+3.2)** | **44.7 (+0.1)** |

*Table 9.* **MT-Bench evaluation results.** We assess multi-turn conversation capabilities on **Gemma-2B-it** and **Llama-3.1-8B-Instruct**. The scores (1st turn, 2nd turn, and Average) are graded by GPT-4. The best Average scores within each iteration group are marked in **bold**.

| METHOD | ITER | PM: GEMMA-2B-IT 1ST | 2ND | AVG | PM: LLAMA-3.1-8B-INSTRUCT 1ST | 2ND | AVG |
|---|---|---|---|---|---|---|---|
| BASE | – | 8.39 | 7.75 | 8.07 | 8.39 | 7.75 | 8.07 |
| BT+SPPO | 1 | 8.38 | 7.62 | 8.00 | 8.42 | 8.08 | **8.25 (+0.14)** |
| GPM+SPPO | 1 | 8.58 | 7.85 | **8.21** | 8.38 | 7.42 | 7.90 |
| HRC+SPPO | 1 | 8.45 | 7.98 | **8.21** | 8.39 | 7.16 | 7.78 |
| HRC+DSPPO | 1 | 8.48 | 7.75 | 8.11 | 8.46 | 7.75 | 8.11 |
| BT+SPPO | 2 | 8.21 | 7.74 | 7.98 | 8.39 | 7.92 | 8.16 |
| GPM+SPPO | 2 | 8.30 | 7.65 | 7.98 | 8.61 | 8.10 | 8.36 |
| HRC+SPPO | 2 | 8.39 | 7.96 | 8.18 | 8.61 | 8.24 | **8.42 (+0.06)** |
| HRC+DSPPO | 2 | 8.49 | 8.10 | **8.29 (+0.11)** | 8.55 | 7.74 | 8.14 |
| BT+SPPO | 3 | 8.48 | 7.69 | 8.08 | 8.21 | 7.78 | 7.99 |
| GPM+SPPO | 3 | 7.98 | 7.42 | 7.70 | 7.85 | 7.25 | 7.55 |
| HRC+SPPO | 3 | 8.50 | 7.72 | 8.11 | 8.48 | 7.94 | **8.21 (+0.22)** |
| HRC+DSPPO | 3 | 8.61 | 7.86 | **8.24 (+0.13)** | 8.04 | 7.68 | 7.86 |

## C.7. Validation of GPT-based Evaluation

Given that our evaluation relies on GPT-family models as judges, we acknowledge the potential concern of judge bias in automated evaluation. To validate the reliability of our GPT-based evaluation pipeline, we conduct three complementary analyses: (1) agreement between GPT evaluation and human annotations, (2) cross-evaluation using multiple GPT variants.

**GPT vs. Human Agreement.** To assess the consistency between GPT-based evaluation and human judgments, we sampled 500 response pairs from AlpacaEval 2.0 and collected annotations from a human evaluator. Each response pair was evaluated by both GPT-4o-mini (using the standard AlpacaEval 2.0 protocol) and a human annotator, who determined which response was preferred. The resulting confusion matrix is shown in Table 10.

*Table 10.* **Agreement between GPT-4o-mini and Human Evaluation** on 500 AlpacaEval 2.0 samples.

|  | HUMAN POSITIVE | HUMAN NEGATIVE | TOTAL |
|---|---|---|---|
| **GPT POSITIVE** | TP = 247 | FP = 32 | 279 |
| **GPT NEGATIVE** | FN = 26 | TN = 195 | 221 |
| **TOTAL** | 273 | 227 | 500 |

The Cohen's kappa coefficient of $\kappa \approx 0.7655$ indicates substantial agreement between GPT-based evaluation and human judgments.

**Cross-Evaluation with Multiple GPT Variants.** To further assess the robustness of our conclusions to the choice of judge, we performed cross-evaluation using multiple GPT-based evaluators. We evaluated the model obtained at the third iteration (trained with the LLaMA-3.1-8B-Instruct preference model) on AlpacaEval 2.0 using different GPT variants: GPT-4o-mini, GPT-4.1, and GPT-5-mini. The results are shown in Table 11.

*Table 11.* **Cross-Evaluation Results using Different GPT-Based Evaluators** on AlpacaEval 2.0 (Iteration 3). All methods use Llama-3.1-8B-Instruct as the preference model. The best scores within each evaluator are highlighted in **bold**. The consistent relative rankings across evaluators demonstrate robustness to judge selection.

| METHOD | GPT-4O-MINI | GPT-4.1 | GPT-5-MINI |
|---|---|---|---|
| BT + SPPO | 40.56 | 35.25 | 43.05 |
| GPM + SPPO | 41.14 | 35.77 | 44.89 |
| **HRC + DSPPO** | **41.90** | **36.14** | **46.11** |

The results show consistent relative rankings across all three evaluators: HRC + DSPPO achieves the highest Length-Controlled Win Rate (LC. WR), followed by GPM + SPPO, then BT + SPPO. This consistency across different GPT variants indicates that our conclusions are robust to the specific choice of GPT-based judge.

**Summary.** Collectively, these analyses provide evidence that our GPT-based evaluation pipeline produces reliable and robust results. The substantial agreement between GPT and human evaluations and the consistent rankings across multiple GPT variants support the validity of our evaluation conclusions. While automated evaluation cannot fully replace human judgment, these validation steps give us confidence that the relative performance differences reported in our main experiments are meaningful and not artifacts of judge bias.

## C.8. Scalability Evaluation on Gemma-2-9B-it

To evaluate whether the performance gains of HRC+DSPPO transfer to stronger backbone models, we conduct a scalability experiment using **Gemma-2-9B-it** (Team et al., 2024b) as the backbone. Due to computational constraints, we use a 4-bit quantized version of Gemma-2-9B-it for efficient inference. Critically, we directly reuse the preference models (BT and HRC, both trained on Gemma-2B-it) without any retraining or fine-tuning on Gemma-2-9B-it data. This setup provides a direct test of whether the preference signals learned by HRC on a smaller model generalize effectively to guide the alignment of a larger, more capable policy.

**Experimental Setup.** Following the same protocol as our main alignment experiments, we perform SPPO and DSPPO over $T = 3$ iterations. We compare three settings: (1) the unaligned Gemma-2-9B-it base model, (2) BT+SPPO as a baseline alignment pipeline, and (3) HRC+DSPPO, our full method. All models are evaluated on AlpacaEval 2.0 using GPT-4o-mini as the judge, with Length-Controlled Win Rate (LC. WR) as the primary metric.

**Results and Analysis.** Table 12 presents the results across iterations. The base Gemma-2-9B-it model achieves an LC. WR of 38.38%. Applying BT+SPPO for three iterations improves performance to 48.79%, yielding a substantial gain of +10.41%. Notably, HRC+DSPPO reaches 42.90% after just a single iteration, already approaching the BT+SPPO peak. By iteration 3, HRC+DSPPO achieves a peak LC. WR of **52.20%**, outperforming BT+SPPO by +3.41% and surpassing the base model by +13.82%. These results demonstrate that the preference signals decomposed by HRC generalize effectively across model scales, enabling stronger alignment outcomes even when the preference model is trained on a smaller backbone. The progressive improvement across DSPPO iterations further confirms that dynamic scheduling of transitive and cyclic signals facilitates stable optimization on larger models.

*Table 12.* **Scalability evaluation on Gemma-2-9B-it (4-bit quantized).** We report the Length-Controlled Win Rate (LC. WR, %) on AlpacaEval 2.0. Preference models (BT, HRC) are trained on Gemma-2B-it and directly applied to post-train Gemma-2-9B-it without retraining. The best score is highlighted in **bold**.

| METHOD | ITERATION | LC. WR (%) |
|---|---|---|
| GEMMA-2-9B-IT (BASE) | – | 38.38 |
| BT + SPPO | 3 | 48.79 |
| HRC + DSPPO | 1 | 42.90 |
| HRC + DSPPO | 2 | 51.74 |
| HRC + DSPPO | 3 | **52.20** |

# D. Response Examples in Different Iterations

We present a representative case study sampled from the AlpacaEval 2.0 benchmark.Table 13 compares the responses generated by base model (Llama-3.1-8B-Instruct) and our method (HRC+DSPPO).

As shown in Table 13, the iterative process yields observable gains in two key dimensions:

- Correction of Factual Hallucinations: The Base Model initially hallucinates the dish's origin as the *Veneto region*. Interestingly, Iteration 1 inherits and even elaborates on this error (adding "Padua"). However, Iteration 2 effectively "unlearns" the false information by adopting a neutral stance, paving the way for Iteration 3 to correctly identify the traditionally accepted origin (*Lombardy/Milan*), thereby achieving factual alignment.
- Structural Refinement: The response structure evolves from unstructured text blocks to a well-organized layout. Iteration 3 demonstrates superior instruction following by using bold keys and categorized lists, making the information more accessible and professional compared to the baseline.

*Table 13.* **Examples on AlpacaEval 2.0.** We compare the responses generated by Llama-3.1-8B-Instruct and our method.

---

**Prompt:** What is Gremolata?

---

**Llama-3.1-8B-Instruct**

Gremolata is a traditional Italian condiment originating from the Veneto region. It's a mixture of lemon zest, garlic, and parsley, often used to add flavor to various dishes, particularly meats and soups. The ingredients are finely minced and combined to create a paste-like consistency. Here's a basic recipe for gremolata: * 1/4 cup freshly grated lemon zest * 4-5 cloves of garlic, minced * 1/4 cup chopped fresh parsley * 2 tablespoons olive oil (optional) Mix all the ingredients together, and adjust to taste. Some recipes may also include a pinch of salt or a squeeze of fresh lemon juice. Gremolata is commonly used as a condiment to: 1. Add flavor to osso buco (braised veal shanks) 2. Enhance the taste of roasted meats, such as beef or pork 3. Add a burst of citrus flavor to soups and stews 4. Use as a sauce for pasta or risotto 5. Add a flavor boost to vegetables, like asparagus or Brussels sprouts Gremolata is a versatile condiment that's easy to make and can elevate the flavors of many dishes.

---

**Iteration 1**

Gremolata is a traditional Italian condiment originating from the Veneto region, particularly associated with the city of Padua. It is a mixture of lemon zest, garlic, and parsley, typically combined in a specific ratio and often served with meat dishes, particularly osso buco (braised veal shanks). The ingredients are finely minced and mixed together, creating a flavorful and aromatic paste. Gremolata is often used as a condiment to add flavor to the dish rather than as an ingredient in the cooking process itself. The name "gremolata" comes from the Venetian dialect, with "grem" meaning "to press" or "to squeeze," likely referring to the process of squeezing the lemon zest into the mixture. Gremolata is a versatile condiment that can be served as a garnish or mixed into sauces, soups, and other dishes to add a bright, citrusy flavor.

---

**Iteration 2**

Gremolata is a classic Italian condiment made from a mixture of lemon zest, garlic, and herbs, typically parsley or rosemary. The ingredients are finely chopped and combined to create a bright, citrusy, and aromatic flavor profile. Traditionally, gremolata is used to add flavor to osso buco, a classic Italian dish made with braised veal shanks. The gremolata is often served on top of the dish, adding a burst of freshness and a hint of citrus to balance the richness of the veal. The name "gremolata" comes from the Italian word "gremolare," which means "to pound or crush," referring to the process of finely chopping the ingredients. Gremolata can also be used as a condiment for other dishes, such as pasta, roasted meats, or vegetables, and is a popular ingredient in many Italian recipes. Here's a simple recipe for homemade gremolata: Ingredients: * 1/4 cup freshly chopped parsley * 2 cloves garlic, minced * 2 tablespoons freshly grated lemon zest * 1 tablespoon olive oil * Salt, to taste Combine the parsley, garlic, lemon zest, and olive oil in a bowl. Mix well and season with salt to taste. Store gremolata in an airtight container in the refrigerator for up to 1 week. Gremolata is a versatile condiment that adds a bright, citrusy flavor to many dishes. Give it a try and enjoy!

---

**Iteration 3**

Gremolata is a classic Italian condiment originating from the Lombardy region, particularly associated with the city of Milan. It's a simple yet flavorful mixture of ingredients that's commonly used to add a burst of citrus flavor to various dishes, particularly osso buco, a traditional braised veal shank dish. The traditional gremolata recipe typically consists of: 1. **Lemon zest**: The grated outer skin of a lemon, which provides a bright, citrusy flavor and aroma. 2. **Garlic**: Minced or finely chopped garlic, which adds a pungent flavor. 3. **Parsley**: Chopped fresh parsley, which adds a fresh, herbaceous note. Some variations may also include additional ingredients, such as: * Olive oil * Salt * Black pepper * Lemon juice (some recipes use a squeeze of fresh lemon juice in addition to the zest) To make gremolata, you simply mix these ingredients together in a bowl, often in a specific proportion. The traditional ratio is about 2 parts lemon zest to 1 part garlic to 1 part parsley, but feel free to adjust to taste. Gremolata is often used as a condiment to: * Enhance the flavor of osso buco * Add a bright, citrusy note to soups, stews, and braises * Top grilled meats, fish, or vegetables for a burst of flavor * Use as a garnish for pasta dishes or risottos In summary, gremolata is a zesty, aromatic, and flavorful condiment that adds a touch of Italian flair to various dishes.

---

