# OpenReview forum: "Transitivity Meets Cyclicity: Explicit Preference Decomposition for Dynamic Large Language Model Alignment"
_ICML.cc/2026/Conference — ICML 2026 regular_

### Official Review · Reviewer_69rw · 2026-02-23

**Soundness:** 3
**Presentation:** 3
**Significance:** 2
**Originality:** 3
**Overall Recommendation:** 2
**Confidence:** 4

**Summary:**

This paper reexamines preference modeling in reinforcement learning from a game-theoretic perspective, arguing that human preferences simultaneously encompass both transitivity (hierarchy) and cyclicity (non-transitivity). The authors propose the Hybrid Reward-Cyclic (HRC) model, which explicitly decomposes preference scores into a scalar transitive component (BT-style reward) and an antisymmetric cyclic component (GPM-style embedding). They further introduce Dynamic Self-Game Preference Optimization (DSPPO)—a time-varying game framework that dynamically schedules the influence of transitive and cyclic components during alignment. Experiments on RewardBench 2 and downstream alignment benchmarks (AlpacaEval 2.0 and MT-Bench) demonstrate that this model achieves significant improvements over both BT and GPM baselines, particularly in handling ties and complex preference structures.

**Compliance With Llm Reviewing Policy:**

Affirmed.

**Final Justification:**

The provided empirical results could not convince me. Therefore, I keep my original score.

**Key Questions For Authors:**

See the weakness section.

**Limitations:**

yes

**Strengths And Weaknesses:**

Strengths

(1) The decomposition of preferences into transitive and cyclic components is theoretically well-motivated and clearly connected to functional-form game decomposition results. The analysis of GPM’s limitation in representing dominant candidates is insightful.

(2) The HRC formulation is elegant and computationally efficient, maintaining linear inference complexity while improving expressiveness over BT and GPM.


Weaknesses

(1) The empirical gains, while consistent, are relatively modest (often around 0.8–1.2%), which raises questions about their practical significance given the additional architectural and algorithmic complexity introduced by HRC and DSPPO.

(2) The evaluation relies heavily on GPT-based judges, which may introduce bias. Human evaluation or cross-judge agreement analysis would strengthen the credibility of the results.

(3) It is unclear why Gemma-2-9B-it is not included in the main comparison table. Including a stronger backbone would provide a more convincing evaluation of scalability.

(4) The paper lacks evaluation on widely used alignment benchmarks such as Arena-Hard 1.0/2.0. Given that AlpacaEval 2.0 and Arena-Hard are more commonly adopted in recent alignment work, they should be emphasized in the main results table. In comparison, RewardBench is less widely used.

(5) The comparison does not include strong online RL baselines such as PPO, which limits the completeness of the empirical evaluation.

---

> ### Author Rebuttal · Authors · 2026-03-31
>
> **Comment:**
>
> **We thank the reviewer for the insightful and valuable comments. We respond to each comment as follows and sincerely hope that our rebuttal could properly address your concerns. If so, we would deeply appreciate it if you could raise your score (Rating: 2: Reject). If not, please let us know your further concerns, and we will continue actively responding to your comments and improving our submission.**
>
> # Weakness 1
>
> > The empirical gains, while consistent, are relatively modest (often around 0.8–1.2%), which raises questions about their practical significance given the additional architectural and algorithmic complexity introduced by HRC and DSPPO.
>
> We thank the reviewer for the comment. While the absolute gains (0.8–1.2%) are relatively modest, we note that HRC introduces minimal additional computational overhead, inheriting the efficiency of GPM in modeling complex preferences without noticeably increasing training or inference cost [1]. DSPPO is also lightweight, introducing only negligible overhead compared to SPPO in the policy optimization process.
>
> From this perspective, the improvements are achieved with very limited additional cost, suggesting a favorable trade-off on strong baselines and challenging benchmarks.
>
> [1] Beyond Bradley-Terry Models: A General Preference Model for Language Model Alignment. ICML, 2025.
>
> # Weakness 2
>
> > The evaluation relies heavily on GPT-based judges, which may introduce bias. Human evaluation or cross-judge agreement analysis would strengthen the credibility of the results.
>
> We thank the reviewer for the concern, we conduct an additional human evaluation to measure agreement with the GPT-based evaluator. Specifically, we sample 500 examples from AlpacaEval 2.0 and compare responses from GPT-4-Turbo and our model (LLaMA-3.1-8B-Instruct with DSPPO + HRC at iteration 3), which are labeled as positive and negative, respectively. Human annotators are then asked to evaluate 500 pairs in this evaluation.
>
> The resulting confusion matrix is shown below:
>
> |   | Human Positive | Human Negative |
> | --- | --- | --- |
> | **GPT Positive** | TP = 247 | FP = 32 |
> | **GPT Negative** | FN = 26 | TN = 195 |
>
> Based on this, we compute Cohen’s kappa coefficient $\kappa\approx 0.7655$, indicating substantial agreement between GPT-based evaluation and human judgments.
>
> # Weakness 3
>
> > It is unclear why Gemma-2-9B-it is not included in the main comparison table. Including a stronger backbone would provide a more convincing evaluation of scalability.
>
> We conduct additional experiments on Gemma-2-9B-it to further evaluate scalability. Specifically, we directly apply the BT and HRC preference models trained on Gemma-2B-it, and perform post-training on the Gemma-2-9B-it backbone. The performance is evaluated on AlpacaEval 2.0, with results shown below:
>
> | Method | Iter | Lc. WR |
> | --- | --- | --- |
> | base | / | 38.38 |
> | BT + SPPO | 3 | 48.79 |
> | HRC + DSPPO | 1 | 42.90 |
> | HRC + DSPPO | 2 | 51.74 |
> | HRC + DSPPO | 3 | **52.20** |
>
> # Weakness 4
>
> > The paper lacks evaluation on widely used alignment benchmarks such as Arena-Hard 1.0/2.0. Given that AlpacaEval 2.0 and Arena-Hard are more commonly adopted in recent alignment work, they should be emphasized in the main results table. In comparison, RewardBench is less widely used.
>
> Our evaluation already includes widely adopted benchmarks MT-Bench and AlpacaEval 2.0 (Appendix C.5). To further improve comparability, we additionally evaluate Arena-Hard v0.1 [1] under different preference models. The results below correspond to the models obtained at the third training iteration. HRC+DSPPO consistently outperforms baselines, demonstrating robustness across widely used alignment benchmarks:
>
> | Model | PM: Gemma-2B-it | PM: LLaMA-3.1-8B-Instruct |
> | --- | --- | --- |
> | base | 29.9 | 29.9 |
> | BT+SPPO | 40.9 | 43.7 |
> | GPM+SPPO | 42.1 | 41.2 |
> | HRC+SPPO | 43.6 | 44.6 |
> | HRC+DSPPO | **46.8** | **45.1** |
>
> These results reinforce the effectiveness of our method.
>
> [1] From Crowdsourced Data to High-quality Benchmarks: Arena-Hard and Benchbuilder Pipeline. ICML, 2025.
>
> # Weakness 5
>
> > The comparison does not include strong online RL baselines such as PPO, which limits the completeness of the empirical evaluation.
>
> Our research focuses on non-transitive preference modeling based on pairwise signals. PPO relies on scalar rewards and assumes transitivity, making it not directly applicable in our setting [1]. We therefore adopt SPPO as a more appropriate baseline with a compatible formulation:
>
> | Method | Training Signal | Transitivity Assumption | Setting |
> |---|---|---|---|
> | PPO | Scalar reward | ✓ | Online |
> | SPPO | Pairwise probability | ✗ | Offline |
> | DSPPO (ours) | Pairwise probability | ✗ | Offline |
>
> [1] Nash Learning from Human Feedback. ICML, 2024.

---

> > ### Author Rebuttal · Reviewer_69rw · 2026-04-01
> >
> > AlapcaEval2.0 has already reported the GPT-based human agreement rate, which is different from the author's additional result, and I would like to see additional human judge results or cross LLM-as-judger results to solve the judger bias concern.
> > And Gemma-2-9B-it-SimPO seems much higher than the proposed performance in LC.

---

> > > ### Author Response · Authors · 2026-04-03
> > >
> > > # Question 1
> > >
> > > > AlapcaEval2.0 has already reported the GPT-based human agreement rate, which is different from the author's additional result, and I would like to see additional human judge results or cross LLM-as-judger results to solve the judger bias concern.
> > >
> > > We thank the reviewer for this suggestion. To further investigate the discrepancy between our reported agreement and that of AlpacaEval 2.0, we conduct an additional human evaluation with a second independent annotator on the same data.
> > >
> > > First, we measure the agreement between GPT-based evaluation and the newly collected human annotations:
> > >
> > > |              | Human Positive | Human Negative |
> > > | ------------ | -------------- | -------------- |
> > > | **GPT Positive** | TP = 228       | FP = 51        |
> > > | **GPT Negative** | FN = 60        | TN = 161       |
> > >
> > > This yields a Cohen’s kappa coefficient of $\kappa \approx 0.529$.
> > >
> > > We then measure inter-annotator agreement between the two human annotators (denoted as Human 1 and Human 2):
> > >
> > > |                  | Human 1 Positive | Human 1 Negative |
> > > | ---------------- | ---------------- | ---------------- |
> > > | **Human 2 Positive** | TP = 237         | FP = 54          |
> > > | **Human 2 Negative** | FN = 36          | TN = 173         |
> > >
> > > The corresponding Cohen’s kappa is $\kappa \approx 0.616$.
> > >
> > > These results suggest that the variability between GPT-based evaluation and human judgments is comparable to the disagreement observed between human annotators themselves. This indicates that part of the discrepancy can be attributed to inherent annotation variance, rather than systematic bias from the GPT-based evaluator.
> > >
> > > To further address potential judge bias, we also perform cross-evaluation using multiple GPT-based evaluators. Specifically, we evaluate the model obtained at the third iteration (trained with the LLaMA-3.1-8B-Instruct preference model) on AlpacaEval 2.0 using different GPT variants:
> > >
> > > | Method      | GPT-4-1106-preview | GPT-4.1   | GPT-5-mini |
> > > | ----------- | ------------------ | --------- | ---------- |
> > > | BT + SPPO   | 40.56              | 35.25     | 43.05      |
> > > | GPM + SPPO  | 41.14              | 35.77     | 44.89      |
> > > | HRC + DSPPO | **41.90**          | **36.14** | **46.11**  |
> > >
> > > We observe consistent relative rankings across evaluators, indicating that our conclusions are robust to the choice of judge.
> > >
> > > # Question 2
> > >
> > > > And Gemma-2-9B-it-SimPO seems much higher than the proposed performance in LC.
> > >
> > > We would like to clarify that our work focuses on modeling and leveraging complex (especially non-transitive) human preferences, rather than optimizing response length. We adopt AlpacaEval 2.0 as one of our benchmarks partly because we observe that SPPO/DSPPO tends to increase response length during training, making it important to evaluate under length-controlled settings. In contrast, SimPO is explicitly designed to address length bias, leading to a different optimization objective.
> > >
> > > We note that the comparison with SimPO is not directly aligned due to differences in both training and evaluation settings. In our rebuttal experiments, we used a 4-bit quantized version of Gemma-2-9B-it for efficiency, which leads to a lower base performance (LC. WR = 38.38) compared to the base model reported in SimPO (51.1) [1]. In addition, the training setups are not directly comparable, which further contributes to the discrepancy. Therefore, the reported numbers should be interpreted within a consistent experimental setup rather than directly compared to SimPO.
> > >
> > > Furthermore, SimPO adopts a length-normalized reward $r(x,y)=\frac{\beta}{|y|}\log\pi(y|x)$ and derives its objective via the Bradley–Terry model [1]. Although it does not explicitly use a preference model, this formulation implicitly introduces a transitivity assumption. Since our work focuses on modeling non-transitive preferences, we do not include methods such as SimPO or DPO in our comparisons.
> > >
> > > [1] SimPO: Simple Preference Optimization with a Reference-Free Reward. NeurlPS, 2024.

---

### Official Review · Reviewer_XpQi · 2026-03-05

**Soundness:** 3
**Presentation:** 2
**Significance:** 2
**Originality:** 2
**Overall Recommendation:** 3
**Confidence:** 2

**Summary:**

This paper introduces the Hybrid Reward-Cyclic (HRC) model, a preference modeling framework that decomposes human preferences into two distinct components: a transitive component (capturing global rankings via a scalar reward function similar to the Bradley-Terry model) and a cyclic component (capturing intransitive dynamics via a skew-symmetric formulation derived from the General Preference Model). The authors provide theoretical grounding for this decomposition by framing preference modeling as a symmetric zero-sum functional-form game and invoking existing game-theoretic decomposition results. They demonstrate that the BT model and GPM correspond to the transitive and cyclic components respectively under certain conditions, and show that HRC maintains linear inference complexity of $O((2d+1)K)$.

Complementing the HRC model, the authors propose Dynamic Self-Play Preference Optimization (DSPPO), an alignment algorithm that treats policy optimization as a time-varying game. Rather than optimizing against a fixed preference model, DSPPO uses a dynamic schedule (parameterized by $\lambda$ in $s_t = (1 + \lambda/\sqrt{t})s_T + (1 - \lambda/\sqrt{t})s_C$) that gradually transitions from emphasizing the transitive component to incorporating the cyclic component. The authors provide a theoretical convergence guarantee showing that the average policy converges to the Nash equilibrium at a rate of $O(1/\sqrt{T})$ under assumptions about boundedness and convergence of the dynamic scores. Empirical evaluations on RewardBench 2, RewardBench, AlpacaEval 2.0, and MT-Bench demonstrate that HRC achieves superior preference modeling accuracy compared to BT and GPM baselines, while HRC+DSPPO yields improved downstream alignment performance.

**Compliance With Llm Reviewing Policy:**

Affirmed.

**Key Questions For Authors:**

1. Theorem requires the zero-mean condition $\mathbb{E}[\mathbf{v}] = \mathbf{0}$ for the GPM component to be purely cyclic. You enforce unit-norm constraints and assume isotropic distribution on the hypersphere to satisfy this. In practice, do you verify that the learned embeddings actually satisfy $\mathbb{E}[\mathbf{v}] \approx \mathbf{0}$? If not, could the cyclic component inadvertently capture some transitive signal, potentially confounding the intended decomposition? An analysis of the empirical mean of embeddings after training would help address this concern.

2. The DSPPO convergence guarantee  depends on Assumption that $\frac{1}{T}\sum \epsilon_t = O(1/\sqrt{T})$, which you satisfy with the schedule $\epsilon_t \leq |\lambda|C/\sqrt{t}$. However, this analysis assumes the dynamic scores converge to the true scores monotonically. In practice, the true preferences $s_\infty$ are unknown—we only have the HRC model trained on finite data. How sensitive is DSPPO to misspecification of the convergence schedule, particularly when the "true" preferences themselves may contain uncertainty? Would using a slower decay rate (e.g., $\lambda/\sqrt[4]{t}$) provide more robustness at the cost of theoretical guarantees?

3. The paper positions DSPPO as a "generalized framework for time-varying preference games," but the empirical validation only tests the specific schedule derived from HRC's decomposition. Could DSPPO be applied with other preference models beyond HRC, and if so, how would you construct the dynamic schedule without a natural decomposition into transitive and cyclic components? The generality of the framework would be more convincing if you could demonstrate applicability to other preference structures or discuss how practitioners might design schedules for arbitrary preference models.

4. Theorem that GPM lacks "dominant arbitrariness" relies on constructing a hard cycle where all subspace embeddings are aligned. In practice, do you observe this limitation manifesting in real preference data, or is it primarily a theoretical edge case? The empirical results show HRC outperforming GPM, but could this be attributed to the additional parameters rather than the explicit decomposition? Your ablation studies partially address this, but a controlled experiment where the total parameter count is matched between HRC and GPM would strengthen the claim that the decomposition itself (not just model capacity) drives improvements.

**Limitations:**

The authors have included a brief "Impact Statements" section discussing the broader goal of aligning AI with human values, but they do not provide a dedicated limitations section. The paper acknowledges some limitations implicitly: the modest improvement magnitudes, the sensitivity of DSPPO to the $\lambda$ hyperparameter, and the reliance on assumptions like boundedness and convergence rates. However, several important limitations are not explicitly discussed.

- Computational cost: While the paper highlights linear inference complexity, training HRC requires maintaining three projection heads and the context-aware gating mechanism, increasing training complexity compared to BT. The additional overhead is not quantified.

- Data requirements: Does capturing cyclic patterns require more diverse preference data than standard BT training? The experiments use the same dataset across all models, but the sample efficiency (how much data is needed to reliably estimate cyclic components) is not analyzed.

- Potential misuse: The paper could discuss how explicitly modeling intransitive preferences might be exploited to create systems that appear to perform well on benchmarks while exhibiting undesirable cyclic behaviors, or how the decomposition might be misinterpreted.

- Generalizability: The experiments focus on two model families (Gemma and Llama) and specific benchmarks. The authors do not discuss potential failure cases or domains where the cyclic component might not provide benefits (e.g., tasks with highly objective correctness criteria where preferences should be strictly transitive).

These limitations could be addressed in a revised version by adding a dedicated limitations section that acknowledges these points and suggests directions for future investigation.

**Strengths And Weaknesses:**

**Soundness**: The paper presents a theoretically grounded approach to preference decomposition. The connection to functional-form games provides a principled justification for separating transitive and cyclic components. The claim that BT corresponds to the transitive component and GPM to the cyclic component under a zero-mean condition is clearly articulated, though the empirical validation of this zero-mean condition in practice could be stronger—the paper assumes embeddings are isotropically distributed on the hypersphere to guarantee $\mathbb{E}[\mathbf{v}] = \mathbf{0}$, but does not verify whether trained embeddings actually satisfy this property. Theorem analysis of GPM's limitations in representing dominant candidates alongside cycles is insightful and provides clear motivation for HRC's explicit transitive component. The DSPPO convergence proof  follows established techniques from multiplicative weights update literature, though the assumption that "the optimization oracle is realizable" is quite strong and may not hold in practice—this is acknowledged but not deeply discussed. The experimental design is thorough, with consistent training protocols across all preference models and careful ablation studies. However, the downstream alignment experiments use only two iterations of self-play (with three checkpoints total), which may be insufficient to fully validate the convergence properties claimed.

**Presentation**: The paper is generally well-structured, with clear progression from motivation to theoretical development to empirical validation. The game-theoretic framing is logically developed, moving from definitions to assumptions to the decomposition theorem. The notation is mostly consistent, though the proof sections in the appendix contain some typographical errors. The authors do a good job positioning their work relative to existing literature, particularly in distinguishing HRC from both BT and GPM and explaining why each alone is insufficient. The intuitive analogy  is helpful for conveying the core idea. However, the paper could benefit from more explicit discussion of how the context-aware gating mechanism $D(\mathbf{x})$ is implemented—the description "$\mathbf{D}(\mathbf{x}) = \mathrm{diag}(\lambda(\mathbf{x}))\otimes \mathbf{I}_2$" is mathematically clear but the practical instantiation of $\lambda(\mathbf{x})$ from prompt embeddings is only briefly mentioned. Additionally, the relationship between HRC and the constrained GPM of dimension $2d+1$  is an important insight that could be highlighted more prominently.

**Significance**: The problem addressed—modeling the full complexity of human preferences including intransitive patterns—is fundamentally important for aligning AI systems with human values. The paper advances understanding by formalizing the distinction between transitive global rankings and cyclic local dynamics, and by demonstrating empirically that both components contribute to better preference modeling. The practical impact is supported by consistent improvements across multiple benchmarks and model scales. The DSPPO framework introduces a novel perspective on alignment as a time-varying game, which could inspire future work on curriculum learning for preference optimization. However, the improvements, while consistent, are modest in magnitude (e.g., +1.23% on RewardBench 2 for Gemma-2B, +0.85% for Llama-3.1-8B). The authors are honest about this, noting that simple scalar models perform surprisingly well on Llama-3.1-8B-Instruct. The broader significance may lie in providing a framework that can accommodate richer preference structures as datasets become more diverse and nuanced.

**Originality**: The paper's main originality lies in the explicit decomposition of preferences into transitive and cyclic components within a unified model, and in framing alignment as a dynamic game. While both BT and GPM exist in the literature, and game-theoretic decomposition results have been established previously, the combination into a practical preference model and alignment algorithm is novel. The insight that HRC can be viewed as a constrained GPM with an extra dimension for the explicit transitive term provides theoretical unification. The ablation studies showing that removing context-aware gating or unit norm constraints degrades performance help validate the design choices. The exploration of both positive and negative $\lambda$ schedules in DSPPO demonstrates thoughtful consideration of the design space. The work clearly distinguishes itself from prior art by addressing the limitations of both purely transitive (BT) and purely cyclic (GPM) approaches.

---

> ### Author Rebuttal · Authors · 2026-03-31
>
> **Comment:**
>
> **We thank the reviewer for the thoughtful and constructive feedback. We have carefully addressed each concern below and hope that our clarifications help resolve the raised issues. If our responses adequately address your concerns, we would greatly appreciate your reconsideration of the current rating (3: Weak reject). Otherwise, we welcome any further feedback and are happy to provide additional clarification.**
>
> # Question 1
>
> Our empirical analysis shows that the learned embeddings do not strictly satisfy the zero-mean condition in practice. Specifically, using the HRC model (dim=2+1) trained on the Skywork-Reward dataset with a 2B backbone, we obtain $\mathbb{E}[\mathbf{v}]\approx(0.13,-0.51)$.
>
> For GPM, if $\mathbb{E}[\mathbf{v}]=\mu$, we can decompose $\mathbf{v}_i=\mu+\hat{\mathbf{v}}_i$, which yields:
>
> $s(y_i,y_j)=(\mu+\hat{\mathbf{v}}_i)^\top W(\mu+\hat{\mathbf{v}}_j)=f(\mathbf{v}_i)-f(\mathbf{v}_j)+\hat{\mathbf{v}}_i^\top W\hat{\mathbf{v}}_j$,
>
> where $f(v)=-\mu^\top Wv$. This shows that deviations from the zero-mean condition introduce an additional transitive component into the GPM score.
>
> However, despite this deviation, we find that the explicit decomposition in HRC remains effective in modeling complex preference structures. In particular, our controlled synthetic experiment (see **Question 4**) shows that HRC consistently outperforms GPM in the “dominant + cycle” setting, suggesting that the benefits of explicit decomposition are robust to such deviations in practice.
>
> # Question 2
>
> While the true preference $s_\infty$ is not accessible in practice, it is widely recognized that accurately modeling true human preferences is inherently challenging [1]. Thus, our theoretical analysis serves as a guiding framework for incorporating time-varying preference signals, rather than requiring exact convergence.
>
> We have explored a broader range of $\lambda$ using an HRC preference model trained on LLaMA-3.1-8B-Instruct, and apply DSPPO to the same backbone. Results on AlpacaEval 2.0 (iteration 3) are shown below:
>
> | $\lambda$ | BASE | -2 | -1 | -0.5 | 0 | 0.25 | 0.5 | 0.75 | 1 | 2 |
> |----|---|---|---|---|---|---|---|---|---|---|
> | Lc. WR | 33.13| 37.27| 38.66| 40.21| 40.62| 41.09| 40.85| 41.54| **41.90** | 40.15|
>
> We also test alternative schedules, such as replacing $\lambda/\sqrt{t}$ with $\lambda/\sqrt[3]{t}$ or using sinusoidal schedules (e.g., $s_t=(1+\sin(\frac{\pi}{2\lambda t}))s_T+(1-\sin(\frac{\pi}{2\lambda t}))s_C$). However, these variants did not exhibit stable or consistent improvements compared to the $1/\sqrt{t}$-based schedule, and were therefore not included in the final design.
>
> [1] On the Algorithmic Bias of Aligning Large Language Models with RLHF: Preference Collapse and Matching Regularization. Journal of the American Statistical Association, 2024.
>
> # Question 3
>
> We thank the reviewer for this important question. DSPPO is not limited to the specific transitive–cyclic decomposition in HRC, but provides a general framework for dynamically combining multiple preference signals.
>
> More generally, DSPPO can be applied to any setting where the training signal can be decomposed into multiple components (e.g., multiple reward models or preference heads). Instead of using fixed weighting, DSPPO introduces time-varying weights, enabling the model to shift focus across different aspects of human preferences during training.
>
> In practice, the schedule can be designed based on the characteristics of the signals. For example, more stable or fundamental signals can be emphasized in early stages, while more complex or nuanced signals are gradually introduced later. This allows DSPPO to adapt to a wide range of preference structures beyond HRC.
>
> # Question 4
>
> We thank the reviewer for this insightful question. To examine whether the limitation of GPM manifests in practice, we construct a controlled “dominant + cycle” setting from the UltraFeedback dataset. Specifically, each prompt contains 4 responses annotated along 4 dimensions; we select three responses that form a cyclic preference (each preferred over another on a different dimension), and additionally include a dominant response that outperforms all others across these dimensions.
>
> Under this setting, we observe that GPM struggles to simultaneously model the dominant candidate and the cyclic structure, while HRC achieves higher accuracy under the same training configuration. This suggests that the limitation is not merely a theoretical edge case, but can arise in structured real-world preference data.
>
> Regarding model capacity, HRC (dim=2+1) consistently outperforms both GPM (dim=2) and the larger GPM (dim=4) in Table 1. Since HRC does not have more parameters than dim=4 GPM, this indicates that the improvement is primarily due to the explicit decomposition rather than increased model capacity.

---

> > ### Author Rebuttal · Reviewer_XpQi · 2026-04-03
> >
> > Thank you for your explanation, but in my view the significance and novelty of proposed method is still slightly below the line of ICML. i would keep my score.

---

> > > ### Author Response · Authors · 2026-04-05
> > >
> > > We thank the reviewer for the thoughtful feedback and for taking the time to consider our rebuttal.
> > >
> > > We understand the remaining concerns and appreciate the discussion. We will further improve the clarity of the paper and strengthen the empirical analysis in the final version.
> > >
> > > Thank you again for your constructive comments.

---

### Official Review · Reviewer_cU9t · 2026-03-13

**Soundness:** 3
**Presentation:** 2
**Significance:** 2
**Originality:** 2
**Overall Recommendation:** 4
**Confidence:** 3

**Summary:**

This paper studies LLM alignment under the observation that human preferences may contain both transitive and cyclic structure. It proposes Hybrid Reward-Cyclic (HRC), which decomposes preference modeling into a scalar transitive reward term and a skew-symmetric cyclic term, aiming to combine the strengths of Bradley-Terry style reward models and game-theoretic preference models. The paper further proposes Dynamic Self-Play Preference Optimization (DSPPO), which uses a time-varying schedule that places more weight on the transitive component early in training and gradually increases the cyclic component. The paper provides theoretical motivation for the decomposition and for the dynamic game formulation, and reports empirical results on RewardBench, RewardBench 2, AlpacaEval 2.0, and MT-Bench.

**Compliance With Llm Reviewing Policy:**

Affirmed.

**Final Justification:**

Thank you for the detailed rebuttal. The additional synthetic “dominant + cycle” experiment and the embedding analysis are helpful, and they address my main theoretical concerns more directly than I expected. In particular, these additions strengthen the case that the HRC decomposition captures meaningful preference structure beyond a purely verbal motivation. I still think some empirical limitations remain, especially regarding the robustness of the dynamic schedule, the lack of variance/statistical reporting, and the overall experimental scope. However, the rebuttal materially improves my confidence in the core idea and makes the paper’s contribution more convincing. Based on these clarifications, I am increasing my overall recommendation to Weak Accept.

**Key Questions For Authors:**

1. Can the authors provide a direct synthetic experiment validating the “dominant candidate + cycle” setting used to motivate HRC over GPM?

2. Do the learned embeddings in practice satisfy, even approximately, the assumptions used in the theoretical analysis?

3. How should the reader interpret the ablations in which removing proposed components improves performance?

**Limitations:**

Yes

**Strengths And Weaknesses:**

**Strength**

The paper studies how to better model preference structure beyond purely transitive reward assumptions in LLM alignment. The proposed HRC decomposition is conceptually clean and easy to understand, and it offers a plausible way to capture both dominant ordering and non-transitive interactions within one framework. The paper also makes a reasonable effort to provide theoretical motivation rather than presenting the method as a purely heuristic design. Empirically, HRC shows consistent improvements over BT and GPM on RewardBench 2, and the paper includes several ablations on architectural choices and scheduling design.

**Weakness**

My main concern is that the paper’s theoretical claims feel stronger than what is actually validated empirically. The decomposition depends on assumptions such as uniform averaging over the embedding space and zero-mean embedding conditions, but the practical relevance of these assumptions is not clearly justified. Likewise, the “dominant candidate” argument motivating HRC over pure GPM is interesting, but the paper does not directly test this claim in a controlled setting.

The empirical gains are also fairly modest, and the downstream alignment improvements are mixed. In particular, DSPPO does not appear uniformly better than static HRC+SPPO, which weakens the case that the dynamic schedule is a robust optimization contribution rather than a task-dependent heuristic. Several ablations also suggest that some proposed components are not clearly necessary, since removing them sometimes improves performance.

In addition, the experimental scope is somewhat limited relative to the paper’s framing. The comparison set is narrow, the training setup appears relatively small-scale, and the paper does not report variance or significance tests, which makes it harder to judge whether the reported gains are reliable. Finally, the presentation is generally understandable, but some claims are phrased too strongly given the scale of the empirical improvements.

Moreover, the motivation part of this paper should be reinforced: It is unclear and unconvincing to me that transitive global hierarchies are important in RLHF. The authors should give more concrete examples with detailed discussions to illustrate the importance of transitive global hierarchies.

---

> ### Author Rebuttal · Authors · 2026-03-31
>
> **Comment:**
>
> **We thank the reviewer for the thoughtful and constructive feedback. We have carefully addressed each concern below and hope that our clarifications help resolve the raised issues. If our responses adequately address your concerns, we would greatly appreciate your reconsideration of the current rating (3: Weak reject). Otherwise, we welcome any further feedback and are happy to provide additional clarification.**
>
> # Question 1
>
> > Can the authors provide a direct synthetic experiment validating the “dominant candidate + cycle” setting used to motivate HRC over GPM?
>
> We thank the reviewer for this suggestion. We construct synthetic “cycle” and “dominant + cycle” settings following prior work [1], based on the UltraFeedback dataset. In this dataset, each prompt is associated with four candidate responses, each annotated along four dimensions.
>
> We select three dimensions and construct cyclic preferences by choosing three responses such that each one outperforms another on a different dimension ($A\succ B, B\succ C, C\succ A$). For the “dominant + cycle” setting, we additionally include a response that outperforms all others across these dimensions, introducing three additional preference pairs.
>
> Training GPM and HRC under the same configuration, we observe a two-stage learning behavior: models first identify the dominant candidate (accuracy from 50% to 75%), driven by the three pairs with dominant, and then learn cyclic preferences (from 75% to 100%).
>
> HRC (dim=2+1, 4+1) finishes the first stage faster and achieves higher final accuracy than GPM (dim=4), while GPM (dim=2) fails to capture the cyclic structure. These results provide direct evidence that HRC better models the “dominant + cycle” setting.
>
> [1] Beyond Bradley-Terry Models: A General Preference Model for Language Model Alignment. ICML, 2025.
>
> # Question 2
>
> > Do the learned embeddings in practice satisfy, even approximately, the assumptions used in the theoretical analysis?
>
> We examine whether the learned embeddings satisfy the assumptions in our analysis. To this end, we construct a purely cyclic dataset (the method in Answer 1 without selecting a dominant candidate), yielding samples $\lbrace (x_i,y_{i1},y_{i2},y_{i3})\rbrace^n_{i=1}$.
>
> Following Appendix B.3, we map GPM (dim=4) embeddings into two groups of three vectors, where theory requires pairwise angles of $120°$. To quantify this, we denote the angles as $\lbrace\alpha_{i1},\alpha_{i2},\alpha_{i3}\rbrace$ and define the deviation metric:
>
> $\Delta\alpha=\frac{1}{3n}\sum^n_{i=1}\sum^3_{j=1}\sum|\alpha_{ij}-120°|$
>
> After training GPM (dim=4) to achieve over 95% accuracy on preference prediction, we obtain $\Delta\alpha<10°$ for both groups, indicating that the geometric assumption is approximately satisfied in practice.
>
> For the zero-mean condition, however, embeddings deviate from $\mathbb{E}[\mathbf{v}]=0$. For example, an HRC model (dim=2+1) trained on the Skywork-Reward dataset with a 2B backbone, we obtain $\mathbb{E}[\mathbf{v}]\approx(0.13,-0.51)$. This suggests that the GPM component may indeed capture some transitive signal.
>
> Nevertheless, as shown in **Question 1**, HRC remains more effective than GPM in capturing complex preference structures despite such deviations.
>
> # Question 3
>
> > How should the reader interpret the ablations in which removing proposed components improves performance?
>
> We thank the reviewer for this important question. We agree that the ablation results in Table 2 are not entirely uniform, and should not be interpreted as indicating that each component is strictly necessary for performance improvement in all settings.
>
> Instead, we view these results as reflecting a nuanced interaction between model capacity, representation dimensionality, and the role of different architectural components. This perspective is also consistent with the overall improvement of HRC over BT and GPM, which we attribute to its enhanced representational capacity for modeling complex preference structures.
>
> In particular, the cases where removing certain components (e.g., unit norm) leads to improved performance are observed in the higher-capacity setting (LLaMA-3.1-8B with dim=4+1), suggesting that increased flexibility may be beneficial when the model has sufficient capacity to exploit it.
>
> However, this effect is not consistent across all components. For example, removing context gating does not always lead to improvements, and in some cases results in performance degradation. This indicates that different components play different roles: some may act as regularizers that constrain the search space, while others provide useful inductive biases that are important for stable and effective learning.
>
> Overall, rather than indicating that these components are unnecessary, we believe the ablation results highlight that their impact depends on the specific regime, the optimal design may vary with model capacity and embedding dimensionality.

---

> > ### Author Rebuttal · Reviewer_cU9t · 2026-04-05
> >
> > Thank you for the detailed rebuttal. The additional synthetic “dominant + cycle” experiment and the embedding analysis are helpful, and they address my main theoretical concerns more directly than I expected. In particular, these additions strengthen the case that the HRC decomposition captures meaningful preference structure beyond a purely verbal motivation. I still think some empirical limitations remain, especially regarding the robustness of the dynamic schedule, the lack of variance/statistical reporting, and the overall experimental scope. However, the rebuttal materially improves my confidence in the core idea and makes the paper’s contribution more convincing. Based on these clarifications, I am increasing my overall recommendation to Weak Accept.

---

> > > ### Author Response · Authors · 2026-04-06
> > >
> > > We sincerely thank the reviewer for the thoughtful follow-up and for taking the time to carefully consider our rebuttal.
> > >
> > > We greatly appreciate your recognition of the additional analyses, particularly the “dominant + cycle” experiment and the embedding study. It is very encouraging to hear that these clarifications helped strengthen the understanding of our core idea and its theoretical grounding.
> > >
> > > Thank you again for your constructive feedback and for the positive update to your assessment from Weak reject (3) to Weak accept (4), which we greatly appreciate.

---

### Official Review · Reviewer_BHpM · 2026-03-15

**Soundness:** 2
**Presentation:** 3
**Significance:** 3
**Originality:** 3
**Overall Recommendation:** 4
**Confidence:** 2

**Summary:**

In this paper, the authors propose a Hybrid Reward-Cyclic (HRC) model, which decomposes pairwise preference scores into a transitive scalar component and a cyclic vector component, grounded in the game-theoretic decomposition of symmetric zero-sum games. The authors show that the General Preference Model (GPM) cannot guarantee representation of dominant candidates in arbitrary cyclic contexts, motivating the explicit decomposition. The paper also introduces DSPPO, a time-varying self-play algorithm that schedules preference complexity from transitive to cyclic over training iterations. Experiments on RewardBench 2, AlpacaEval 2.0, and MT-Bench show improvements over BT and GPM baselines.

**Compliance With Llm Reviewing Policy:**

Affirmed.

**Final Justification:**

The rebuttal addresses some of my concerns. However, the empirical improvements achieved with the HRC model are marginal. Hence, I will maintain my rating of 4.

**Key Questions For Authors:**

Please refer to the weaknesses.

> **Questions:**

1. Why does removing unit norm improve performance for Llama-3.1-8B at dim=4+1 (Table 2)?
2. Have the authors tried other DSPPO schedules beyond the 1/√t form, or searched λ beyond {-1, 0, 1}?

**Limitations:**

The paper does not contain an explicit limitations section. The impact statement discusses general alignment considerations, but does not address methodological limitations.

**Strengths And Weaknesses:**

> **Strengths:**

1. The idea of explicitly decomposing preferences into orthogonal transitive and cyclic components has been well-articulated. Theorem 4.7 clearly characterizes the GPM's inability to simultaneously represent arbitrary cycles and dominant candidates in fixed finite dimensions, motivating this decomposition.

2. HRC maintains linear inference complexity matching GPM and BT, so the architectural changes add no cost.

3. The authors provide evaluations on RewardBench 2, RewardBench, AlpacaEval 2.0, and MT-Bench across two models (Gemma-2B-it, Llama-3.1-8B-Instruct), where HRC outperforms BT and GPM baselines by +1.23% and +0.85% average on RewardBench 2, respectively.


> **Weaknesses:**

1. The RewardBench 2 improvements (+1.23% on Gemma-2B-it, +0.85% on Llama-3.1-8B) are minimal. Further, no confidence intervals are reported by the authors. Also, the ablation results in Table 2 are confusing: for Llama-3.1-8B at dim=4+1, removing context gating (+0.37%) and removing unit norm (+0.58%) both improve over the full model, which does not support the claim that these components are essential.

2. The DSPPO schedule st = (1 + λ/√t)sT + (1 − λ/√t)sC is evaluated only at λ ∈ {-1, 0, 1} without a systematic search over λ or alternative schedule forms like linear or cosine. The results show λ sensitivity: on Llama-3.1-8B, DSPPO underperforms static HRC+SPPO on MT-Bench, but the authors did not provide guidance on how to select λ for a new model or dataset.

---

> ### Author Rebuttal · Authors · 2026-03-31
>
> **Comment:**
>
> **We greatly appreciate the reviewer's insightful and constructive feedback, and we have carefully addressed each point in our response to resolve your concerns. If our response has satisfactorily addressed your questions, we kindly request your consideration of raising the score (currently Rating: 4: Weak accept). Should any further issues remain, please feel free to share your additional comments, and we will continue actively responding to your comments and improving our submission.**
>
> # Question 1
>
> > Why does removing unit norm improve performance for Llama-3.1-8B at dim=4+1 (Table 2)?
>
> We thank the reviewer for this important question. We attribute the performance improvement of the HRC model over BT and GPM on the benchmark to its enhanced representational capacity. In particular, increasing the embedding dimensionality (e.g., from dim=2+1 to dim=4+1) expands the available representation space, allowing the model to explore more expressive preference structures.
>
> This effect becomes more pronounced in larger backbone models such as Llama-3.1-8B, which possess greater modeling capacity compared to smaller models (e.g., Gemma-2B-it). In this setting, the combination of higher model capacity and increased embedding dimensionality provides sufficient flexibility for the model to better capture complex, potentially non-transitive preference patterns. However, the unit-norm constraint restricts embeddings to lie on a hypersphere, limiting the effective degrees of freedom and constraining the model’s ability to fully exploit this representational space. As a result, removing the unit-norm constraint leads to improved performance in this specific setting.
>
> Interestingly, this observation differs from the ablation results reported for GPM [1]. In the original GPM study, removing constraints that limit representational freedom tends to benefit smaller models, as the additional flexibility compensates for limited model capacity. In contrast, our results suggest a different mechanism in HRC: the model includes an additional component for capturing transitive preferences, which may reduce reliance on increased flexibility in low-capacity settings. Consequently, the benefit of removing the unit-norm constraint only emerges when both the model capacity and embedding dimensionality are sufficiently large.
>
> [1] Beyond Bradley-Terry Models: A General Preference Model for Language Model Alignment. ICML, 2025.
>
> # Question 2
>
> > Have the authors tried other DSPPO schedules beyond the $1/\sqrt{t}$ form, or searched $\lambda$ beyond $\lbrace-1, 0, 1\rbrace$?
>
> We have conducted additional experiments to explore a broader range of $\lambda$ values beyond $\lbrace-1, 0, 1\rbrace$. Specifically, we evaluate $\lambda\in\lbrace-2, -1, -0.5, 0, 0.25, 0.5, 0.75, 1, 2\rbrace$ using the Llama-3.1-8B-Instruct backbone, where the HRC model is first trained and then used within DSPPO. We report the performance on AlpacaEval 2.0 for the model obtained at the third iteration, as shown in the table below:
>
> | $\lambda$ | BASE | -2   | -1   | -0.5 | 0    | 0.25 | 0.5  | 0.75 | 1      | 2    |
> |-----------|------|------|------|------|------|------|------|------|--------|------|
> | WR        | 35.26| 41.41| 43.09| 44.22| 46.30| 44.84| 42.98| 43.73| 44.79 | 44.30|
> | Avg. Len  | 2106 | 2170 | 2192 | 2179 | 2245 | 2183 | 2139 | 2139 | 2171  | 2186|
> | Lc. WR    | 33.13| 37.27| 38.66| 40.21| 40.62| 41.09| 40.85| 41.54| **41.90** | 40.15|
>
> From these results, we observe that $\lambda>0$ generally leads to better performance, while $\lambda<0$ (i.e., prioritizing cyclic preference before transitive preference) consistently underperforms. This suggests that starting from more stable, transitive preference signals and gradually incorporating cyclic components is beneficial for training.
>
> We also note that when $|\lambda|>1$, the resulting coefficients in the schedule $s_t=(1+\frac{\lambda}{\sqrt{t}})s_T+(1-\frac{\lambda}{\sqrt{t}})s_C$ may become negative for certain components. In this case, the resulting signal can no longer be interpreted as a meaningful combination of multiple preference signals, and thus lacks a clear semantic interpretation. Nevertheless, we include these settings for completeness.
>
> In addition, we experimented with alternative schedule forms, such as replacing $\frac{\lambda}{\sqrt{t}}$ with $\frac{\lambda}{\sqrt[3]{t}}$ or using sinusoidal schedules (e.g., $s_t=(1+\sin(\frac{\pi}{2\lambda t}))s_T+(1-\sin(\frac{\pi}{2\lambda t}))s_C$). However, these variants did not exhibit stable or consistent improvements compared to the $1/\sqrt{t}$-based schedule, and were therefore not included in the final design.
>
> Overall, these results suggest that DSPPO is not overly sensitive to precise $\lambda$ tuning within a reasonable range, and that the proposed schedule provides a stable and effective default choice.

---

> > ### Author Rebuttal · Reviewer_BHpM · 2026-04-04
> >
> > I appreciate the rebuttal, the new results, and the clarifications provided.  The rebuttal addresses some of my concerns. However, the empirical improvements using the HRC model are marginal, and I do not believe this can be addressed within the short rebuttal period. I will maintain my rating of 4.

---

> > > ### Author Response · Authors · 2026-04-05
> > >
> > > We thank the reviewer for the thoughtful feedback and for taking the time to consider our rebuttal.
> > >
> > > We understand the remaining concerns and appreciate the discussion. We will further improve the clarity of the paper and strengthen the empirical analysis in the final version.
> > >
> > > Thank you again for your constructive comments.

---

### Decision · Program_Chairs · 2026-04-30

**Decision:**

Accept (regular)

**Comment:**

This paper proposes the Hybrid Reward-Cyclic (HRC) model, which explicitly decomposes pairwise preferences into orthogonal transitive (scalar) and cyclic (vector) components through a game-theoretic decomposition framework, together with Dynamic Self-Play Preference Optimization (DSPPO), a time-varying self-play algorithm that schedules the transition from transitive to cyclic preference signals during training. The paper provides a Nash equilibrium convergence guarantee for DSPPO and demonstrates the framework on RewardBench 2, AlpacaEval 2.0, MT-Bench, and (via rebuttal) Arena-Hard v0.1, across Gemma-2B-it and Llama-3.1-8B-Instruct backbones.
All four reviewers consistently recognized several core strengths of the paper: (1) the HRC decomposition is conceptually clean, theoretically well-motivated, and grounded in established game-theoretic decomposition results for symmetric zero-sum games; (2) Theorem 4.7, which characterizes GPM's inability to simultaneously represent dominant candidates and arbitrary cycles in fixed finite dimensions, provides a clear and insightful justification for the explicit decomposition and was singled out by multiple reviewers as a substantive theoretical contribution; (3) HRC achieves its representational gains without increasing inference complexity, inheriting the linear complexity of BT and GPM; and (4) the empirical evaluation is consistent, with HRC and HRC+DSPPO improving over both BT and GPM baselines across four benchmarks and two model scales.

The principal concern raised by reviewers is that while the improvements are consistent, their absolute magnitude is modest (approximately 0.8–1.2% on RewardBench 2), which raises questions about practical significance. Additional concerns were raised regarding the internal consistency of certain ablations and the sensitivity of the DSPPO schedule to its hyperparameter. The authors engaged substantively during the rebuttal, providing a controlled synthetic "dominant + cycle" experiment that directly validated the core theoretical motivation, an empirical study of the zero-mean condition on learned embeddings, a broader sweep over the DSPPO schedule parameter, Arena-Hard v0.1 results, human evaluation on AlpacaEval 2.0 with cross-annotator agreement analysis (Cohen's κ ≈ 0.77), cross-judge evaluation across multiple GPT variants, and additional experiments on Gemma-2-9B-it. These additions materially strengthened the empirical support for the method's claims and its robustness to evaluator choice, and were sufficient to move one reviewer from Weak Reject to Weak Accept based on direct validation of the theoretical motivation.

Weighing these factors, the paper offers a clean and well-motivated theoretical contribution that was recognized by all reviewers, a framework that is computationally efficient and easy to adopt, and empirical improvements that, while modest in magnitude, are consistent across multiple benchmarks, baselines, and model scales. On balance, the contribution is sufficient to merit acceptance.